# First-trimester exposure to macrolides and risk of major congenital malformations compared with amoxicillin: A French nationwide cohort study

**Anh Tran** [1,2*], **Mahmoud Zureik**[2], **Jeanne Sibiude**[1,2,3], **Sara Miranda**[2], **Jérôme Drouin**[2], **Lise Marty**[2], **Alain Weill**[2], **Rosemary Dray-Spira**[2], **Xavier Duval**[1,4‡], **Sarah Tubiana**[1,2,4‡]

**1** Université Paris Cité et Université Sorbonne Paris Nord, IAME, INSERM, Paris, France, **2** EPI-PHARE, epidemiology of health products (French National Agency for the Safety of Medicines and Health Products, and French National Health Insurance), Saint-Denis, France, **3** Gynecology-Obstetrics Department, Louis Mourier Hospital, AP-HP, Colombes, France, **4** INSERM CIC1425, Bichat Hospital, AP-HP, Paris, France

‡ These authors share the last authorship on this work.
* anh.tran@ansm.sante.fr

## Abstract

### Background

While macrolides are among the frequently prescribed antibiotics for pregnant women, evidence of their fetal safety remains conflicting. This study aimed to evaluate the risk of major congenital malformations (MCM) after first-trimester exposure to macrolides compared with amoxicillin, focusing on specific MCM subtypes.

### Methods and findings

This nationwide cohort study used data from the Mother-Child EPI-MERES Register nested in the French Health Data System (SNDS). Pregnancies linked with their single-ton live-born infants from January 1, 2010, and December 31, 2020, were included. The macrolide exposure group comprised pregnancies with one or more prescriptions filled for systemic macrolides (erythromycin, spiramycin, roxithromycin, josamycin, clarithromycin, and azithromycin) during the first trimester. The comparator group comprised pregnancies exposed to amoxicillin during the first trimester. Adjusted relative risks (aRR) and 95% CI were estimated by log-binomial regression for any MCM overall and individual MCMs with a prevalence of at least one per 10,000 live-born infants in the macrolide exposure group. Among 7,644,579 eligible pregnancies, 140,708 exposed to macrolides and 592,652 exposed to amoxicillin were included. After adjustment for measured confounders, macrolide exposure during the first trimester was not associated with any MCM overall (aRR 1.00, 95% CI 0.96 to 1.05) compared with amoxicillin. Specifically, no increased risk was found for most individual MCMs. However, an increase in the risk for spina bifida (aRR 1.82, 95% CI 1.22 to 2.71) and syndactyly (aRR 1.65, 95% CI 1.06 to 2.58) was observed. The adjusted risk difference per 10,000 live-born infants was 1.15 (95% CI 0.26 to 2.05) for spina bifida and 0.87 (95% CI 0.01 to 1.72) for syndactyly. Sensitivity analyses consistently

**Data availability statement:** According to data protection and French regulations, the authors cannot publicly release the data from the SNDS. However, any individual or organization (public or private; for-profit or non-profit), can request access to anonymized SNDS data to conduct research or evaluations of public interest upon authorization from the National Commission for Data Protection and Liberties (CNIL), via the French Health Data Hub (https://www.health-data-hub.fr/page/faq-english under the section "About the data available through the Health Data Hub," or by contacting hdh@health-data-hub.fr).

**Funding:** The author(s) received no specific funding for this work.

**Competing interests:** The authors have declared that no competing interests exist.

**Abbreviations:** CI, confidence interval; EUROCAT, European Surveillance of Congenital Anomalies; GA, gestational age; ICD-10, International Classification of Disease, 10th revision; LMP, last menstrual period; MCM, major congenital malformation; PS, propensity score; RD, risk difference; RR, relative risk; SNDS, French National Health System

yielded elevated point estimates for these two MCMs, despite wide confidence intervals and small numbers of events. Residual confounding by indication is possible.

## Conclusions

The findings indicate that macrolide exposure during the first trimester is not strongly associated with an increased risk for most individual MCMs, which is reassuring. However, an increased risk of spina bifida and syndactyly remains possible. Future studies are required to investigate these observations further as evidence continues to grow.

## Author summary
### Why was this study done?

- Macrolides are a frequently prescribed class of antibiotics during pregnancy. However, evidence on the risk of major congenital malformations (MCM) associated with first-trimester macrolide exposure remains inconclusive.
- Previous studies lacked sufficient pregnancies to assess the risk of individual MCMs.

### What did the researchers do and find?

- This nationwide cohort study compared the risk of MCMs overall and 42 individual MCMs among over 140,000 pregnancies exposed to macrolides to over 592,000 pregnancies exposed to amoxicillin (a commonly used antibiotic during pregnancy with a well-established safety profile) during the first trimester.
- There was no association between first-trimester macrolide exposure and the risk of MCMs overall. Specifically, macrolide exposure was not associated with an increased risk of most individual MCMs. However, an increased risk of spina bifida and syndactyly was observed with a small risk difference.

### What do these findings mean?

- The association with spina bifida and syndactyly requires further investigation.
- Macrolides should only be prescribed during the first trimester of pregnancy when necessary, and alternatives with a better-established safety profile should be preferred when possible.

## Introduction

Macrolides are a common antibiotic class to treat bacterial infections during pregnancy [1,2]. In France, over 5% of all pregnancies were exposed to at least one macrolide in 2019 [3]. In the United States, 4.4% of pregnancies ending in live births from commercial insurance beneficiaries were exposed to azithromycin, a macrolide antibiotic, during the first trimester from 2011 to 2020 [4].

Unlike penicillin, which has a well-established record of fetal safety, data for macrolides are less conclusive [5,6]. The risk of major congenital malformations (MCM) associated with macrolide exposure during pregnancy is of typical concern, as macrolides can cross

the placenta, even at a low rate [7]. Although each MCM is distinctly rare, they collectively represent a great burden on global public health as major contributors to neonatal morbidity and mortality [8,9]. Several epidemiological studies have examined the risk of MCMs following macrolide exposure during the first trimester; however, the findings remain conflicting [2,10–13]. A cohort study conducted by Fan and colleagues in 2020 reported an increased risk of cardiovascular malformations after first-trimester macrolide exposure [10]. The latest meta-analysis by Keskin-Arslan and colleagues, which included 18 cohort and case–control studies involving 21,541 infants exposed to macrolides, did not suggest an increased risk of MCMs following macrolide exposure except for a small increase in risk following azithromycin exposure [14]. Indeed, previous studies could not examine the risk of individual MCMs due to insufficient numbers of included pregnancies. Nevertheless, to improve the availability of information for clinical decisions and avoid missing potential teratogenic effects, it is important to study individual MCMs and drugs, which usually require very large cohort studies.

Using data from the French National Health System (SNDS), one of the largest databases of its kind, this study aimed to evaluate the association between first-trimester exposure to macrolides and the risk of MCMs, with a specific focus on individual MCMs, compared with amoxicillin.

## Methods

### Data sources

This study was based on the Mother-Child EPI-MERES Register nested in the SNDS. The SNDS contains comprehensive health insurance reimbursement and hospitalization data for over 99% of the French population. Details about the SNDS are provided in the S1 Text. The EPI-MERES register, which was developed by the EPI-PHARE team, contains the identification number of the mother linked with the identification number of their infants, the first day of the last menstrual period (LMP), date of conception, date of pregnancy outcome, gestational age, and socio-demographic characteristics of the mother. The mothers were linked with their infants using a deterministic algorithm, with a linkage rate exceeding 95%. The date of conception was calculated by subtracting the gestational age (GA), primarily available in the hospitalization data, from the pregnancy outcome date. However, data for same-sex twins are excluded as we cannot distinguish between the twins in the dataset. This register has been described and used to study drug safety during pregnancy in previous studies [15–17]. This study followed the Strengthening the Reporting of Observational Studies in Epidemiology (STROBE) reporting guideline (S1 Checklist).

### Ethics committee approval

The EPI-PHARE team has permanent regulatory access to the data from the SNDS. This study thus did not require specific authorization from the French data protection authority (CNIL). Informed consent was waived as the data were anonymized. This study was approved by the scientific committee of the EPI-PHARE (under the reference T-2023-04-512).

### Study population

We identified pregnancies from women aged 15–49 years living in French territories (excluding Mayotte) who had at least one record in the SNDS in the two preceding years (to ensure reliable exposure and covariate assessments) and linked with their singleton live-born infants in the EPI-MERES for the years 2010–2020. We excluded pregnancies having infants

identified with a chromosomal abnormality, those exposed to teratogenic infections (e.g., toxoplasmosis, cytomegalovirus), and those exposed to known teratogenic drugs during the first trimester. The algorithm used to identify pregnancies exposed to teratogenic infections and a list of known teratogenic drugs are presented in S1 and S2 Tables. Pregnancies with three or more prescriptions filled for spiramycin between conception and delivery were assumed to have a maternal toxoplasmosis infection in our algorithms and excluded.

## Exposure and comparator group

We studied six systemic macrolides (ATC code J01FA) commercialized in France during the study period, including erythromycin, spiramycin, roxithromycin, josamycin, clarithromycin, and azithromycin. The exposure to macrolides was defined as filling one or more prescriptions for any systemic macrolide in outpatient settings during the first trimester, spanning from the first day of LMP to the end of the 12th gestational week. This period encompasses the critical organogenesis phase, between the 5th and 10th gestational weeks, when the risk of MCMs is highest [18]. The comparator group comprised pregnancies exposed to amoxicillin (ATC code J01CA04) during the first trimester to reduce the risk of confounding by indication. To avoid the exposure misclassification, we excluded pregnancies exposed to both amoxicillin and macrolides during the first trimester. Pregnancies were then classified into six individual macrolide exposure groups within the overall macrolide exposure group. Pregnancies in each of the individual macrolide groups were allowed to be exposed to any other macrolide.

## Outcomes

Non-chromosomal MCMs were defined according to the classification system of the European Surveillance of Congenital Anomalies (EUROCAT) Guideline [19]. The presence of MCMs was identified using algorithms based on inpatient ICD-10 International Classification of Disease, 10th revision (ICD-10) diagnostic codes, relevant surgery, or medical procedures in the records of live-born infants up to 1 year after delivery (or 2 years for epispadias, hypospadias, severe microcephaly). The follow-up data were available through 2022. Details of the algorithms, including those for chromosomal anomalies, are provided in S3 Table. The algorithms have been previously applied in the study by Blotière and colleagues (2019) [16], which successfully confirmed the association between valproate and the risk of MCMs.

## Covariates

Potential confounders and their proxies included age at conception, delivery calendar year, socio-demographics, pregnancy-related healthcare utilization, lifestyle factors, pre-existing medical conditions, and healthcare burden within 6 months before pregnancy. The GA at which treatment with macrolides or amoxicillin was initiated was also included. A list of covariates and their definitions are presented in S4 Table.

## Statistical analysis

Baseline covariates were compared for pregnancies exposed to macrolides overall/each of the six macrolides and those exposed to amoxicillin, using standardized differences. For the deprivation index, missing data were treated as a separate category, while no missing data were observed for other covariates. The crude number of affected live-born infants and prevalence per 10,000 live-born infants in all exposure groups were also calculated for any MCM overall and all individual MCMs. Propensity score (PS) fine stratification weighting was applied to

control the confounding [20]. We estimated a separate PS for each macrolide exposure group using a logistic regression model that included covariates listed above. Observations from non-overlapping regions of the PS distribution were trimmed. Because of trimming, 1 pregnancy was excluded from any macrolide/josamycin group, 2 from azithromycin/clarithromycin/erythromycin group, and no pregnancies excluded from spiramycin/roxithromycin group. We created 50 strata based on the distribution of pregnancies exposed to macrolides/each of the six macrolides. The pregnancies exposed to amoxicillin were weighted using the distribution of those exposed to macrolides/each of the six macrolides among the PS strata, and the covariate balance was reassessed. Crude and adjusted relative risks (RR) with 95% confidence intervals (CI) were estimated using log-binomial regression for any MCM overall and individual MCMs with a prevalence of $\geq$ 1 per 10,000 infants from pregnancies exposed to macrolides. To account for the correlation between multiple pregnancies in the same woman, generalized estimating equations were applied. Additionally, risk differences (RD) and their 95% CIs were estimated for the group exposed to macrolides overall.

We conducted analyses for the main macrolide exposure group and then repeated them for each of the six individual macrolide exposure groups as subgroup analyses. The subgroup analyses allowed us to verify whether any increased risk observed in the main analysis persists across the macrolide class.

**Sensitivity analysis.**  We conducted several sensitivity analyses to test the robustness of our results. First, we restricted our analysis to pregnancies exposed to a single antibiotic agent (monotherapy) during the first trimester to address potential confounding by infection severity. Second, we redefined the exposure time window as between the 5th and 12th week after the LMP to avoid the dilution of any risk. Third, we conducted a negative control analysis by using the exposure time window between the 20th and 32nd week after the LMP, a period outside the etiologically relevant window for MCMs. In the absence of association in this window, any observed association in the main analysis is unlikely due to residual confounding.

**Supplementary and post-hoc analyses.**  For supplementary analysis, we included pregnancies exposed to cephalosporin antibiotics during the first trimester and those not exposed to antibiotics from one month before pregnancy through the end of the first trimester as the second and third comparator groups, respectively. We also examined the risk of organ-specific MCM groups to compare with previous studies, though no formal interpretation was provided. Given the protective effect of folic acid against neural tube defects, we assessed any observed association with these MCMs among pregnancies with reimbursed folic acid supplementation. For post-hoc analysis, stratified analyses were conducted based on certain baseline characteristics to determine whether the associations observed in the main analysis remained consistent across different subgroups. Additionally, we applied PS overlap weighting as an alternative method for fine stratification PS in the main analysis to evaluate the sensitivity of results to weighting methods. Finally, since the study population was limited to live births, we estimated the crude RR of non-live births among pregnancies exposed to macrolides compared with amoxicillin in the EPI-MERES register. If macrolide exposure is not strongly associated with non-live births, any observed association for high mortality MCMs is unlikely to be biased [21].

To avoid overlooking potential safety signals that may warrant further investigation, no adjustments were made for multiple comparisons. Instead, we opted to interpret the results based on the association strength, the confidence interval width, the consistency in the results of sensitivity analyses, and the null findings in the negative control analysis. This practice has been adopted in previous studies [22,23]. However, to provide transparency for readers, an additional analysis was performed using the Benjamini–Hochberg correction with a 5% false

discovery rate to account for multiple comparisons in the main analysis [24]. Analyses were performed using SAS Enterprise Guide version 9.4 (SAS Institute, North Carolina, USA).

## Results

### Cohort characteristics

Among 7,644,579 eligible pregnancies linked with their singleton live-born infants, we included 140,708 exposed to macrolides overall and 592,652 exposed to amoxicillin during the first trimester. The completed study flow chart is presented in S1 Fig. Most pregnancies included were exposed to one antibiotic agent and filled only one prescription for the antibiotic of interest during the first trimester (S5 Table). Among the macrolide exposure group, pregnancies were exposed most frequently to azithromycin ($n = 42,585$; 30.3%), followed by spiramycin ($n = 35,259$; 25.1%), clarithromycin ($n = 21,527$; 15.3%), josamycin ($n = 19,100$; 13.6%), roxithromycin ($n = 18,257$; 13.0%), and erythromycin ($n = 7,462$; 5.3%). Pregnancies exposed to macrolides overall were more likely to be among women having assisted reproduction and have lower mean GA at the time of first exposure to the antibiotic of interest compared to pregnancies exposed to amoxicillin (Table 1 and S6 Table). The two groups were similar in terms of other baseline characteristics. After PS weighting adjustment, baseline characteristics were well balanced between macrolides/each of the six individual macrolide exposure groups and the amoxicillin exposure group with a standardized difference of less than 10%.

The crude numbers of affected live-born infants and prevalence per 10,000 infants for 68 individual MCMs of EUROCAT classification in each exposure group are presented in Table 2. There were 2,432 (172.8 per 10,000) and 10,176 (171.7 per 10,000) live-born infants identified with at least one MCM from pregnancies exposed to macrolides overall and amoxicillin, respectively. Heart defects, especially atrial septal and atrioventricular septal defects, accounted for approximately a third of affected infants in each exposure group.

### Relative risks of individual MCMs

The crude and adjusted RRs were estimated for any MCM overall and 42 individual MCMs with a prevalence of ≥ 1 per 10,000 infants exposed to macrolides in utero (Figs 1–4). Macrolide exposure during the first trimester was not associated with any MCM overall (aRR 1.00, 95% CI 0.96 to 1.05) compared with amoxicillin. Regarding individual MCMs, we observed increased risks of spina bifida (aRR 1.82, 95% CI 1.22 to 2.71) and syndactyly (aRR 1.65, 95% CI 1.06 to 2.58) compared with amoxicillin, based on 36 and 31 affected infants in the macrolide exposure group, respectively. The adjusted RD per 10,000 live-born infants was 1.15 (95% CI 0.26 to 2.05) for spina bifida and 0.87 (95% CI 0.01 to 1.72) for syndactyly (S7 Table). Conversely, no meaningful increase in the risk of other individual MCMs was observed with aRRs ranging from 0.76 (95%CI 0.42 to 1.37) for pulmonary valve atresia, to 1.46 (95%CI 0.85 to 2.49) for situs inversus. The aRRs across sensitivity analyses were generally consistent with those of the main analysis (Figs 1–4 and S8 Table). The elevated aRRs for spina bifida and syndactyly persisted across sensitivity analyses, with spina bifida showing a significant increase in the monotherapy analysis (aRR 2.02, 95% CI 1.31 to 3.01) and syndactyly in the narrower exposure window analysis (aRR 2.38, 95% CI 1.42 to 4.01). In the negative control analysis, we did not observe any increased risks of spina bifida and syndactyly associated with macrolide exposure during gestational weeks 20–32.

In individual macrolide analyses, aRRs for the most common MCMs were generally close to the null (S9 Table). In contrast, several elevated point estimates were observed for the less common MCMs. Most were imprecise as there were fewer events in the individual macrolide

**Table 1. Selected characteristics of pregnancies exposed to amoxicillin/macrolides overall/each of the six individual macrolides during the first trimester (before propensity score adjustment).**

| Characteristics | Amoxicillin (N=592,652) | Macrolides (N=140,708) | Azithromycin (N=42,585) | Spiramycin (N=35,259) | Clarithromycin (N=21,527) | Josamycin (N=19,100) | Roxithromycin (N=18,257) | Erythromycin (N=7,462) |
|---|---|---|---|---|---|---|---|---|
| GA at the first exposure, week | 6.4 (3.4) | 5.2 (3.6) | 4 (3.3) | 7.5 (2.9) | 3.6 (3) | 6.3 (3.5) | 4.2 (3.4) | 6.9 (3.3) |
| **Socio-demographic** | | | | | | | | |
| Maternal age, year | 29.4 (5.3) | 29.5 (5.5) | 29.6 (5.8) | 29.3 (5.2) | 29.5 (5.4) | 29.5 (5.3) | 29.8 (5.7) | 29.4 (5.4) |
| Complementary Healthcare Coverage, % | 79,855 (13.5) | 17,275 (12.3) | 6,003 (14.1) | 3,501 (9.9) | 2,763 (12.8) | 2,314 (12.1) | 2,353 (12.9) | 832 (11.1) |
| Deprivation index, quintiles, % | | | | | | | | |
| Q1 (least deprived) | 102,246 (17.3) | 25,786 (18.3) | 8,705 (20.4) | 5,646 (16) | 3,513 (16.3) | 3,586 (18.8) | 3,328 (18.2) | 1,624 (21.8) |
| Q2 | 108,980 (18.4) | 25,245 (17.9) | 7,869 (18.5) | 6,232 (17.7) | 3,911 (18.2) | 3,463 (18.1) | 2,969 (16.3) | 1,421 (19) |
| Q3 | 110,525 (18.6) | 25,512 (18.1) | 7,822 (18.4) | 6,399 (18.1) | 4,111 (19.1) | 3,373 (17.7) | 3,194 (17.5) | 1,250 (16.8) |
| Q4 | 113,008 (19.1) | 26,259 (18.7) | 7,462 (17.5) | 7,029 (19.9) | 4,150 (19.3) | 3,477 (18.2) | 3,540 (19.4) | 1,244 (16.7) |
| Q5 (most deprived) | 133,712 (22.6) | 30,340 (21.6) | 8,064 (18.9) | 8,585 (24.3) | 4,899 (22.8) | 4,317 (22.6) | 3,899 (21.4) | 1,368 (18.3) |
| Missing | 24,181 (4.1) | 7,566 (5.4) | 2,663 (6.3) | 1,368 (3.9) | 943 (4.4) | 884 (4.6) | 1,327 (7.3) | 555 (7.4) |
| **Pregnancy-related healthcare utilization** | | | | | | | | |
| Reimbursed folic acid supplementation, % | 245,114 (41.4) | 57,439 (40.8) | 19,112 (44.9) | 13,944 (39.5) | 7,480 (34.7) | 7,818 (40.9) | 7,183 (39.3) | 3,416 (45.8) |
| Assisted reproduction, % | 14,072 (2.4) | 5,931 (4.2) | 2,643 (6.2) | 897 (2.5) | 238 (1.1) | 455 (2.4) | 1,566 (8.6) | 225 (3) |
| **Lifestyle factors** | | | | | | | | |
| Smoking-related conditions, % | 65,754 (11.1) | 15,548 (11) | 4,579 (10.8) | 4,066 (11.5) | 2,567 (11.9) | 2,093 (11) | 2,016 (11) | 676 (9.1) |
| Alcohol-related conditions, % | 4,243 (0.7) | 1,125 (0.8) | 366 (0.9) | 249 (0.7) | 175 (0.8) | 135 (0.7) | 163 (0.9) | 66 (0.9) |
| Substance use disorders, % | 3,316 (0.6) | 770 (0.5) | 254 (0.6) | 156 (0.4) | 148 (0.7) | 89 (0.5) | 110 (0.6) | 33 (0.4) |
| **Proxies for pre-existing comorbidities** | | | | | | | | |
| Antihypertensive drug use, % | 8,908 (1.5) | 2,232 (1.6) | 664 (1.6) | 520 (1.5) | 361 (1.7) | 307 (1.6) | 338 (1.9) | 120 (1.6) |
| Obesity-related hospital discharge or long-term disease diagnoses, % | 38,186 (6.4) | 8,446 (6) | 2,558 (6) | 1,990 (5.6) | 1,411 (6.6) | 1,231 (6.4) | 1,152 (6.3) | 353 (4.7) |
| Antidiabetic drug use or diabetes-related hospital discharge/long-term disease diagnoses, % | 5,314 (0.9) | 1,196 (0.8) | 420 (1) | 240 (0.7) | 166 (0.8) | 173 (0.9) | 160 (0.9) | 74 (1) |
| **Healthcare burden before pregnancy** | | | | | | | | |
| Prior hospitalization, % | 18,811 (3.2) | 4,769 (3.4) | 1,468 (3.4) | 1,131 (3.2) | 727 (3.4) | 638 (3.3) | 671 (3.7) | 271 (3.6) |
| No. of consultations with general practitioners | 3.9 (3.5) | 4 (3.6) | 4.1 (3.8) | 3.7 (3.4) | 4.3 (3.8) | 4.1 (3.6) | 4.3 (3.8) | 4.2 (3.7) |
| No. of prescribed drugs not antibiotics | 6.2 (5.3) | 6.5 (5.5) | 6.6 (5.5) | 5.9 (5.2) | 6.9 (5.7) | 6.6 (5.5) | 7 (5.6) | 6.5 (5.4) |

*Calendar year and region of residence are not shown.*

exposure groups than in the main analysis. Estimates for spina bifida and syndactyly were consistently elevated in most subgroup analyses, except for certain associations (spiramcyin/erythromycin with spina bifida, erythromycin with syndactyly). The aRRs of spina bifida ranged from 1.58 (95% CI 0.80 to 3.10) for azithromycin to 2.81 (95% CI 1.28 to 6.16) for roxithromycin. For syndactyly, aRRs ranged from 1.54 (95% CI 0.60 to 3.99) for clarithromycin to 2.23 (95% CI 0.90 to 5.54) for josamycin. Sensitivity analyses of the individual macrolide analyses also yielded similar results, especially those of the most common MCMs (S10 Table).

**Supplementary and post-hoc analyses.** The RR estimates for spina bifida and syndactyly following macrolide exposure, compared to cephalosporin exposure, were attenuated but still elevated, particularly for spina bifida (S11 Table). Compared to the unexposed group, the estimates for these two MCMs were similar to those in the main analysis (S12 Table). Results from analyses with organ-specific MCM groups are provided in S13 Table. The aRR for spina bifida was 0.92 (95% CI 0.40 to 2.10) based on seven macrolide-exposed events among

**Table 2. Crude number of affected live-born infants and prevalence (per 10,000) of MCMs, regardless of infant's sex, among pregnancies exposed to amoxicillin/ macrolides overall/ each of the six macrolides during the first trimester (before propensity score adjustment).**

| | Amoxicillin | Macrolides | Azithro-mycin | Spiramycin | Clarithro-mycin | Josamycin | Roxithro-mycin | Erythro-mycin |
|---|---|---|---|---|---|---|---|---|
| **Outcomes** | (*n* = 592,652) | (*n* = 140,708) | (*n* = 42,585) | (*n* = 35,259) | (*n* = 21,527) | (*n* = 19,100) | (*n* = 18,257) | (*n* = 7,462) |
| **Any MCM overall** | 10,176 (171.7) | 2,432 (172.8) | 755 (177.3) | 598 (169.6) | 356 (165.4) | 328 (171.7) | 332 (181.8) | 130 (174.2) |
| **Nervous system anomalies** | 573 (9.7) | 155 (11.0) | 52 (12.2) | 37 (10.5) | 24 (11.1) | 18 (9.4) | 18 (9.9) | 11 (14.7) |
| *Agenesis of the corpus callosum* | 126 (2.1) | 24 (1.7) | 9 (2.1) | 8 (2.3) | 3 (1.4) | 1 (0.5) | 1 (0.5) | 2 (2.7) |
| *Severe microcephaly* | 215 (3.6) | 52 (3.7) | 19 (4.5) | 10 (2.8) | 8 (3.7) | 7 (3.7) | 6 (3.3) | 4 (5.4) |
| *Hydrocephaly* | 131 (2.2) | 43 (3.1) | 13 (3.1) | 15 (4.3) | 6 (2.8) | 4 (2.1) | 4 (2.2) | 3 (4.0) |
| *Spina bifida* | 86 (1.5) | 36 (2.6) | 10 (2.3) | 6 (1.7) | 6 (2.8) | 7 (3.7) | 7 (3.8) | 1 (1.3) |
| Rhinencephaly/Holoprosencephaly | 6 (0.1) | 3 (0.2) | 1 (0.2) | 0 (0) | 1 (0.5) | 1 (0.5) | 0 (0) | 0 (0) |
| Anencephaly | 7 (0.1) | 1 (0.1) | 1 (0.2) | 0 (0) | 0 (0) | 0 (0) | 0 (0) | 0 (0) |
| Encephalocele and meningocele | 22 (0.4) | 4 (0.3) | 1 (0.2) | 1 (0.3) | 0 (0) | 0 (0) | 0 (0) | 2 (2.7) |
| **Eye anomalies** | 156 (2.6) | 36 (2.6) | 10 (2.3) | 10 (2.8) | 2 (0.9) | 7 (3.7) | 6 (3.3) | 1 (1.3) |
| Congenital glaucoma | 57 (1.0) | 12 (0.9) | 5 (1.2) | 2 (0.6) | 1 (0.5) | 2 (1.0) | 1 (0.5) | 1 (1.3) |
| *Congenital cataract* | 84 (1.4) | 20 (1.4) | 3 (0.7) | 8 (2.3) | 2 (0.9) | 3 (1.6) | 4 (2.2) | 0 (0) |
| Cystic eyeball/Other anophthalmos | 4 (0.1) | 1 (0.1) | 1 (0.2) | 0 (0) | 0 (0) | 0 (0) | 0 (0) | 0 (0) |
| Cystic eyeball/Other anophthalmos/Microphthalmos | 37 (0.6) | 10 (0.7) | 4 (0.9) | 3 (0.9) | 0 (0) | 2 (1.0) | 1 (0.5) | 0 (0) |
| **Ear, face, and neck anomalies** | | | | | | | | |
| Anotia and atresia | 34 (0.6) | 8 (0.6) | 3 (0.7) | 3 (0.9) | 0 (0) | 1 (0.5) | 1 (0.5) | 0 (0) |
| **Heart defects** | 3,306 (55.8) | 809 (57.5) | 250 (58.7) | 195 (55.3) | 114 (53.0) | 115 (60.2) | 114 (62.4) | 49 (65.7) |
| *Ventricular septal defect* | 97 (1.6) | 25 (1.8) | 12 (2.8) | 3 (0.9) | 3 (1.4) | 4 (2.1) | 5 (2.7) | 1 (1.3) |
| *Atrial septal defect* | 1,305 (22.0) | 289 (20.5) | 85 (20.0) | 67 (19.0) | 45 (20.9) | 41 (21.5) | 44 (24.1) | 14 (18.8) |
| *Atrioventricular septal* | 1,715 (28.9) | 441 (31.3) | 142 (33.3) | 103 (29.2) | 63 (29.3) | 62 (32.5) | 57 (31.2) | 28 (37.5) |
| *PDA as only CHD in term infants* | 108 (1.8) | 22 (1.6) | 6 (1.4) | 3 (0.9) | 2 (0.9) | 2 (1.0) | 7 (3.8) | 2 (2.7) |
| *Coarctation of aorta* | 227 (3.8) | 48 (3.4) | 14 (3.3) | 15 (4.3) | 8 (3.7) | 9 (4.7) | 3 (1.6) | 6 (8.0) |
| *Hypoplastic left heart* | 79 (1.3) | 22 (1.6) | 7 (1.6) | 7 (2.0) | 2 (0.9) | 3 (1.6) | 3 (1.6) | 1 (1.3) |
| *Aortic valve atresia/stenosis* | 58 (1.0) | 18 (1.3) | 7 (1.6) | 6 (1.7) | 1 (0.5) | 2 (1.0) | 2 (1.1) | 0 (0) |
| *Pulmonary valve stenosis* | 196 (3.3) | 58 (4.1) | 14 (3.3) | 18 (5.1) | 11 (5.1) | 5 (2.6) | 8 (4.4) | 3 (4.0) |
| *Tetralogy of Fallot* | 181 (3.1) | 42 (3.0) | 9 (2.1) | 10 (2.8) | 6 (2.8) | 7 (3.7) | 6 (3.3) | 4 (5.4) |
| *Complete transposition of great arteries (D-TGA)* | 179 (3.0) | 51 (3.6) | 13 (3.1) | 15 (4.3) | 5 (2.3) | 7 (3.7) | 10 (5.5) | 4 (5.4) |
| *Double outlet right ventricle* | 54 (0.9) | 16 (1.1) | 2 (0.5) | 5 (1.4) | 1 (0.5) | 4 (2.1) | 2 (1.1) | 2 (2.7) |
| Anomalous pulmonary venous return | 47 (0.8) | 10 (0.7) | 2 (0.5) | 2 (0.6) | 1 (0.5) | 1 (0.5) | 2 (1.1) | 2 (2.7) |
| Aortic atresia | 24 (0.4) | 8 (0.6) | 4 (0.9) | 2 (0.6) | 0 (0) | 0 (0) | 2 (1.1) | 0 (0) |
| Hypoplastic right heart | 30 (0.5) | 5 (0.4) | 1 (0.2) | 2 (0.6) | 2 (0.9) | 0 (0) | 0 (0) | 0 (0) |
| *Mitral valve atresia/stenosis* | 9 (0.2) | 2 (0.1) | 1 (0.2) | 0 (0) | 1 (0.5) | 0 (0) | 0 (0) | 0 (0) |
| *Pulmonary valve atresia* | 71 (1.2) | 14 (1.0) | 6 (1.4) | 5 (1.4) | 3 (1.4) | 0 (0) | 0 (0) | 0 (0) |
| Ebstein's anomaly | 20 (0.3) | 6 (0.4) | 1 (0.2) | 2 (0.6) | 1 (0.5) | 1 (0.5) | 1 (0.5) | 0 (0) |
| Tricuspid stenosis and atresia | 26 (0.4) | 6 (0.4) | 1 (0.2) | 3 (0.9) | 2 (0.9) | 0 (0) | 0 (0) | 0 (0) |
| Single ventricle | 39 (0.7) | 7 (0.5) | 2 (0.5) | 4 (1.1) | 1 (0.5) | 0 (0) | 1 (0.5) | 0 (0) |
| Corrected transposition of great arteries (L-TGA) | 4 (0.1) | 1 (0.1) | 0 (0) | 0 (0) | 1 (0.5) | 0 (0) | 0 (0) | 0 (0) |
| Double outlet left ventricle | 2 (0) | 2 (0.1) | 0 (0) | 1 (0.3) | 0 (0) | 0 (0) | 1 (0.5) | 0 (0) |
| Common arterial truncus | 25 (0.4) | 4 (0.3) | 2 (0.5) | 0 (0) | 1 (0.5) | 0 (0) | 1 (0.5) | 0 (0) |
| **Respiratory anomalies** | | | | | | | | |
| Choanal stenosis or atresia | 36 (0.6) | 12 (0.9) | 5 (1.2) | 2 (0.6) | 1 (0.5) | 2 (1.0) | 1 (0.5) | 1 (1.3) |
| **Oro-facial clefts** | 821 (13.9) | 161 (11.4) | 46 (10.8) | 42 (11.9) | 30 (13.9) | 23 (12.0) | 14 (7.7) | 9 (12.1) |
| *Cleft lip with or without cleft palate* | 507 (8.6) | 102 (7.2) | 28 (6.6) | 27 (7.7) | 18 (8.4) | 16 (8.4) | 9 (4.9) | 7 (9.4) |
| *Cleft palate* | 314 (5.3) | 59 (4.2) | 18 (4.2) | 15 (4.3) | 12 (5.6) | 7 (3.7) | 5 (2.7) | 2 (2.7) |

*(Continued)*

**Table 2.** (Continued)

| Outcomes | Amoxicillin (n = 592,652) | Macrolides (n = 140,708) | Azithro-mycin (n = 42,585) | Spiramycin (n = 35,259) | Clarithro-mycin (n = 21,527) | Josamycin (n = 19,100) | Roxithro-mycin (n = 18,257) | Erythro-mycin (n = 7,462) |
|---|---|---|---|---|---|---|---|---|
| **Digestive system** | 666 (11.2) | 152 (10.8) | 50 (11.7) | 40 (11.3) | 18 (8.4) | 18 (9.4) | 24 (13.1) | 6 (8.0) |
| *Diaphragmatic hernia* | 122 (2.1) | 22 (1.6) | 7 (1.6) | 5 (1.4) | 4 (1.9) | 3 (1.6) | 3 (1.6) | 1 (1.3) |
| Annular pancreas | 4 (0.1) | 0 (0) | 0 (0) | 0 (0) | 0 (0) | 0 (0) | 0 (0) | 0 (0) |
| Atresia of bile ducts | 29 (0.5) | 5 (0.4) | 4 (0.9) | 1 (0.3) | 0 (0) | 0 (0) | 0 (0) | 0 (0) |
| *Anomalies of intestinal fixation* | 54 (0.9) | 14 (1.0) | 5 (1.2) | 5 (1.4) | 0 (0) | 0 (0) | 2 (1.1) | 2 (2.7) |
| *Hirschsprung's disease* | 66 (1.1) | 18 (1.3) | 6 (1.4) | 5 (1.4) | 2 (0.9) | 2 (1.0) | 4 (2.2) | 0 (0) |
| *Ano-rectal atresia or stenosis* | 168 (2.8) | 46 (3.3) | 13 (3.1) | 10 (2.8) | 9 (4.2) | 6 (3.1) | 6 (3.3) | 2 (2.7) |
| *Atresia or stenosis of intestine* | 66 (1.1) | 17 (1.2) | 7 (1.6) | 5 (1.4) | 0 (0) | 3 (1.6) | 2 (1.1) | 1 (1.3) |
| *Duodenal atresia* | 63 (1.1) | 7 (0.5) | 1 (0.2) | 1 (0.3) | 1 (0.5) | 2 (1.0) | 1 (0.5) | 1 (1.3) |
| *Esophageal atresia* | 137 (2.3) | 34 (2.4) | 14 (3.3) | 9 (2.6) | 2 (0.9) | 4 (2.1) | 6 (3.3) | 0 (0) |
| **Abdominal wall defects** | 147 (2.5) | 37 (2.6) | 9 (2.1) | 9 (2.6) | 8 (3.7) | 4 (2.1) | 5 (2.7) | 3 (4.0) |
| *Omphalocele* | 92 (1.6) | 21 (1.5) | 5 (1.2) | 7 (2.0) | 2 (0.9) | 2 (1.0) | 4 (2.2) | 1 (1.3) |
| *Gastroschisis* | 64 (1.1) | 17 (1.2) | 5 (1.2) | 2 (0.6) | 6 (2.8) | 2 (1.0) | 1 (0.5) | 2 (2.7) |
| **Anomalies of kidney and urinary tract** | 1,539 (26.0) | 331 (23.5) | 91 (21.4) | 103 (29.2) | 38 (17.7) | 47 (24.6) | 46 (25.2) | 17 (22.8) |
| *Lobulated, fused, horseshoe kidney* | 146 (2.5) | 26 (1.8) | 8 (1.9) | 10 (2.8) | 2 (0.9) | 5 (2.6) | 2 (1.1) | 2 (2.7) |
| *Hydronephrosis* | 985 (16.6) | 209 (14.9) | 59 (13.9) | 65 (18.4) | 28 (13.0) | 28 (14.7) | 23 (12.6) | 12 (16.0) |
| *Multicystic renal dysplasia* | 172 (2.9) | 40 (2.8) | 12 (2.8) | 9 (2.6) | 3 (1.4) | 7 (3.7) | 8 (4.4) | 2 (2.7) |
| *Unilateral renal agenesis* | 223 (3.8) | 47 (3.3) | 9 (2.1) | 18 (5.1) | 3 (1.4) | 4 (2.1) | 12 (6.6) | 2 (2.7) |
| Prune Belly syndrome | 5 (0.1) | 2 (0.1) | 0 (0) | 1 (0.3) | 1 (0.5) | 0 (0) | 0 (0) | 0 (0) |
| *Posterior urethral valve* | 67 (1.1) | 18 (1.3) | 5 (1.2) | 5 (1.4) | 2 (0.9) | 2 (1) | 4 (2.2) | 0 (0) |
| Epispadias/Exstrophy of urinary bladder | 15 (0.3) | 4 (0.3) | 1 (0.2) | 2 (0.6) | 0 (0) | 2 (1) | 0 (0) | 0 (0) |
| Bilateral renal agenesis | 13 (0.2) | 0 (0) | 0 (0) | 0 (0) | 0 (0) | 0 (0) | 0 (0) | 0 (0) |
| **Genital anomalies** | 1,468 (24.8) | 355 (25.2) | 113 (26.5) | 78 (22.1) | 64 (29.7) | 36 (18.8) | 54 (29.6) | 16 (21.4) |
| Indeterminate sex | 53 (0.9) | 9 (0.6) | 1 (0.2) | 3 (0.9) | 3 (1.4) | 1 (0.5) | 0 (0) | 1 (1.3) |
| *Hypospadias* | 1,437 (24.2) | 346 (24.6) | 112 (26.3) | 75 (21.3) | 61 (28.3) | 35 (18.3) | 54 (29.6) | 15 (20.1) |
| **Limb anomalies** | 1,641 (27.7) | 414 (29.4) | 133 (31.2) | 89 (25.2) | 59 (27.4) | 69 (36.1) | 57 (31.2) | 18 (24.1) |
| *Syndactyly* | 67 (1.1) | 31 (2.2) | 9 (2.1) | 9 (2.6) | 5 (2.3) | 5 (2.6) | 5 (2.7) | 1 (1.3) |
| *Polydactyly* | 522 (8.8) | 127 (9.0) | 46 (10.8) | 18 (5.1) | 17 (7.9) | 26 (13.6) | 18 (9.9) | 6 (8.0) |
| *Hip dislocation* | 378 (6.4) | 109 (7.7) | 31 (7.3) | 30 (8.5) | 14 (6.5) | 15 (7.9) | 17 (9.3) | 4 (5.4) |
| *Limb reduction* | 148 (2.5) | 27 (1.9) | 9 (2.1) | 8 (2.3) | 3 (1.4) | 5 (2.6) | 1 (0.5) | 1 (1.3) |
| *Club foot – talipes equinovarus* | 559 (9.4) | 131 (9.3) | 42 (9.9) | 27 (7.7) | 21 (9.8) | 19 (9.9) | 18 (9.9) | 6 (8.0) |
| **Other anomalies** | 545 (9.2) | 126 (9.0) | 44 (10.3) | 29 (8.2) | 19 (8.8) | 16 (8.4) | 13 (7.1) | 10 (13.4) |
| *Situs inversus* | 57 (1.0) | 18 (1.3) | 4 (0.9) | 6 (1.7) | 0 (0) | 3 (1.6) | 4 (2.2) | 1 (1.3) |
| *Vascular disruption anomalies* | 183 (3.1) | 43 (3.1) | 15 (3.5) | 9 (2.6) | 9 (4.2) | 5 (2.6) | 3 (1.6) | 4 (5.4) |
| Septo-optic dysplasia | 13 (0.2) | 2 (0.1) | 1 (0.2) | 0 (0) | 1 (0.5) | 0 (0) | 0 (0) | 0 (0) |
| *Craniosynostosis* | 254 (4.3) | 55 (3.9) | 23 (5.4) | 12 (3.4) | 7 (3.3) | 8 (4.2) | 4 (2.2) | 4 (5.4) |
| *Laterality anomalies* | 98 (1.7) | 27 (1.9) | 6 (1.4) | 8 (2.3) | 2 (0.9) | 3 (1.6) | 6 (3.3) | 2 (2.7) |

Outcomes in *italics* are the MCMs with a prevalence of at least one per 10,000 infants among pregnancies exposed to any macrolide.

pregnancies with reimbursed folic acid supplementation. For post-hoc analysis, the aRRs for spina bifida and syndactyly remained elevated across subgroups (S14 Table). In certain strata, the aRRs were toward the null with the limited number of exposed events. When applying PS overlap weights, the results were similar to the main analysis (S15 Table). The crude RR for non-live births among pregnancies exposed to macrolides during the first trimester compared

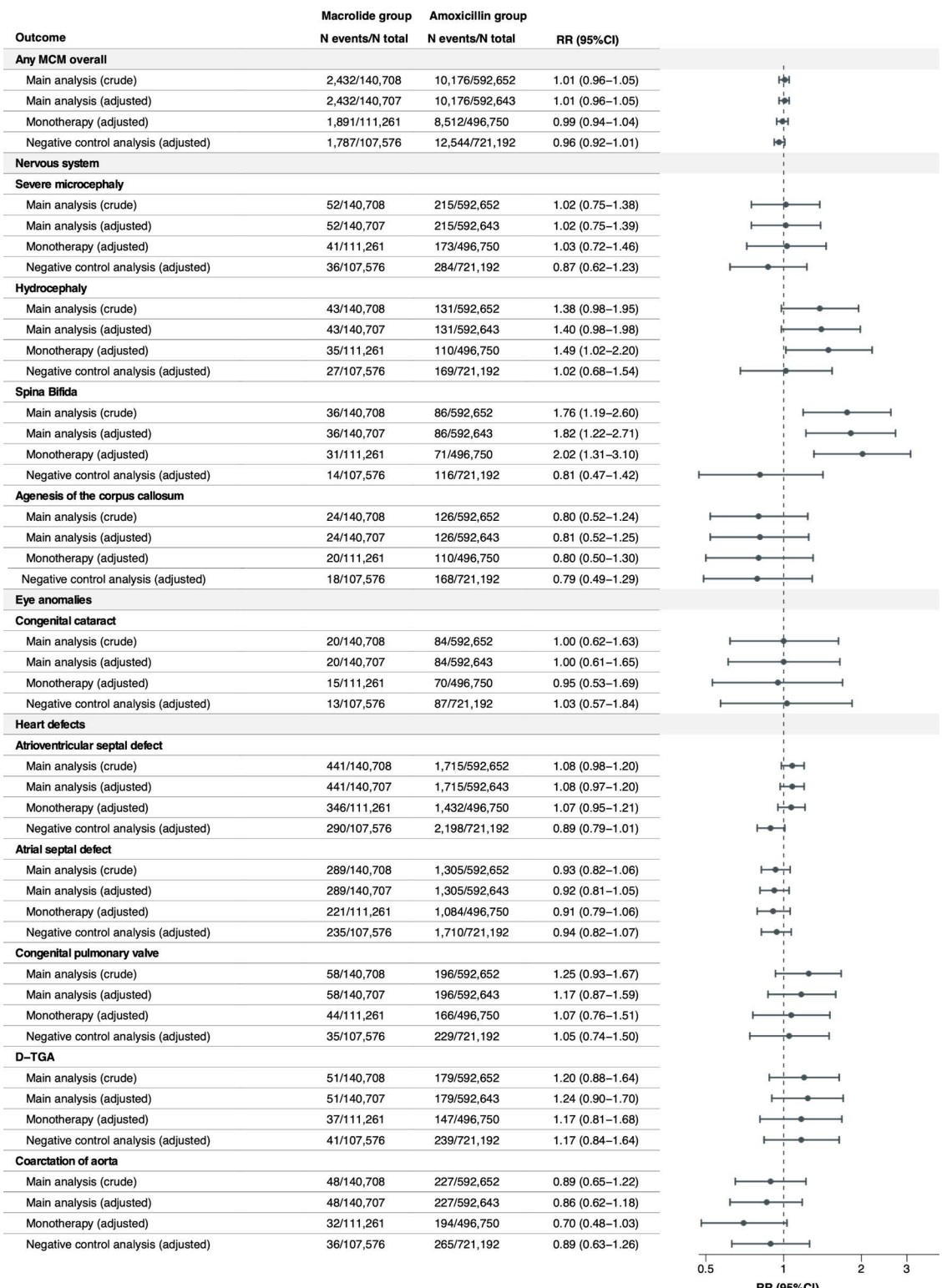

**Fig 1. Relative risks of any MCM and 42 selected individual MCMs (sorted by the most common to the least common MCM in the organ-specific groups) in the macrolide exposure group compared with the amoxicillin exposure group.**

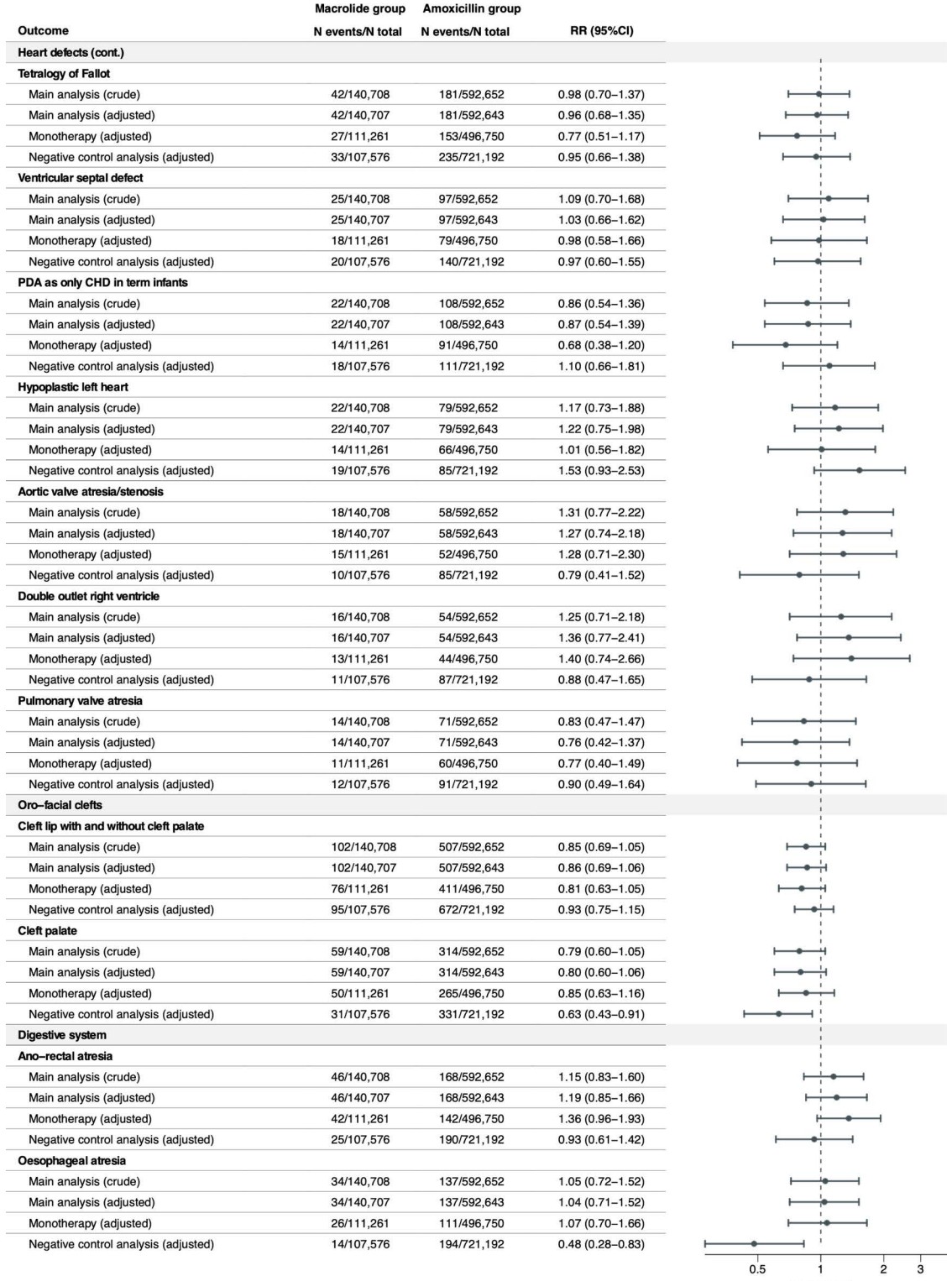

First-trimester macrolide exposure and risk of major congenital malformations

| Outcome | Macrolide group N events/N total | Amoxicillin group N events/N total | RR (95%CI) |
|---|---|---|---|
| **Heart defects (cont.)** | | | |
| **Tetralogy of Fallot** | | | |
| Main analysis (crude) | 42/140,708 | 181/592,652 | 0.98 (0.70–1.37) |
| Main analysis (adjusted) | 42/140,707 | 181/592,643 | 0.96 (0.68–1.35) |
| Monotherapy (adjusted) | 27/111,261 | 153/496,750 | 0.77 (0.51–1.17) |
| Negative control analysis (adjusted) | 33/107,576 | 235/721,192 | 0.95 (0.66–1.38) |
| **Ventricular septal defect** | | | |
| Main analysis (crude) | 25/140,708 | 97/592,652 | 1.09 (0.70–1.68) |
| Main analysis (adjusted) | 25/140,707 | 97/592,643 | 1.03 (0.66–1.62) |
| Monotherapy (adjusted) | 18/111,261 | 79/496,750 | 0.98 (0.58–1.66) |
| Negative control analysis (adjusted) | 20/107,576 | 140/721,192 | 0.97 (0.60–1.55) |
| **PDA as only CHD in term infants** | | | |
| Main analysis (crude) | 22/140,708 | 108/592,652 | 0.86 (0.54–1.36) |
| Main analysis (adjusted) | 22/140,707 | 108/592,643 | 0.87 (0.54–1.39) |
| Monotherapy (adjusted) | 14/111,261 | 91/496,750 | 0.68 (0.38–1.20) |
| Negative control analysis (adjusted) | 18/107,576 | 111/721,192 | 1.10 (0.66–1.81) |
| **Hypoplastic left heart** | | | |
| Main analysis (crude) | 22/140,708 | 79/592,652 | 1.17 (0.73–1.88) |
| Main analysis (adjusted) | 22/140,707 | 79/592,643 | 1.22 (0.75–1.98) |
| Monotherapy (adjusted) | 14/111,261 | 66/496,750 | 1.01 (0.56–1.82) |
| Negative control analysis (adjusted) | 19/107,576 | 85/721,192 | 1.53 (0.93–2.53) |
| **Aortic valve atresia/stenosis** | | | |
| Main analysis (crude) | 18/140,708 | 58/592,652 | 1.31 (0.77–2.22) |
| Main analysis (adjusted) | 18/140,707 | 58/592,643 | 1.27 (0.74–2.18) |
| Monotherapy (adjusted) | 15/111,261 | 52/496,750 | 1.28 (0.71–2.30) |
| Negative control analysis (adjusted) | 10/107,576 | 85/721,192 | 0.79 (0.41–1.52) |
| **Double outlet right ventricle** | | | |
| Main analysis (crude) | 16/140,708 | 54/592,652 | 1.25 (0.71–2.18) |
| Main analysis (adjusted) | 16/140,707 | 54/592,643 | 1.36 (0.77–2.41) |
| Monotherapy (adjusted) | 13/111,261 | 44/496,750 | 1.40 (0.74–2.66) |
| Negative control analysis (adjusted) | 11/107,576 | 87/721,192 | 0.88 (0.47–1.65) |
| **Pulmonary valve atresia** | | | |
| Main analysis (crude) | 14/140,708 | 71/592,652 | 0.83 (0.47–1.47) |
| Main analysis (adjusted) | 14/140,707 | 71/592,643 | 0.76 (0.42–1.37) |
| Monotherapy (adjusted) | 11/111,261 | 60/496,750 | 0.77 (0.40–1.49) |
| Negative control analysis (adjusted) | 12/107,576 | 91/721,192 | 0.90 (0.49–1.64) |
| **Oro–facial clefts** | | | |
| **Cleft lip with and without cleft palate** | | | |
| Main analysis (crude) | 102/140,708 | 507/592,652 | 0.85 (0.69–1.05) |
| Main analysis (adjusted) | 102/140,707 | 507/592,643 | 0.86 (0.69–1.06) |
| Monotherapy (adjusted) | 76/111,261 | 411/496,750 | 0.81 (0.63–1.05) |
| Negative control analysis (adjusted) | 95/107,576 | 672/721,192 | 0.93 (0.75–1.15) |
| **Cleft palate** | | | |
| Main analysis (crude) | 59/140,708 | 314/592,652 | 0.79 (0.60–1.05) |
| Main analysis (adjusted) | 59/140,707 | 314/592,643 | 0.80 (0.60–1.06) |
| Monotherapy (adjusted) | 50/111,261 | 265/496,750 | 0.85 (0.63–1.16) |
| Negative control analysis (adjusted) | 31/107,576 | 331/721,192 | 0.63 (0.43–0.91) |
| **Digestive system** | | | |
| **Ano–rectal atresia** | | | |
| Main analysis (crude) | 46/140,708 | 168/592,652 | 1.15 (0.83–1.60) |
| Main analysis (adjusted) | 46/140,707 | 168/592,643 | 1.19 (0.85–1.66) |
| Monotherapy (adjusted) | 42/111,261 | 142/496,750 | 1.36 (0.96–1.93) |
| Negative control analysis (adjusted) | 25/107,576 | 190/721,192 | 0.93 (0.61–1.42) |
| **Oesophageal atresia** | | | |
| Main analysis (crude) | 34/140,708 | 137/592,652 | 1.05 (0.72–1.52) |
| Main analysis (adjusted) | 34/140,707 | 137/592,643 | 1.04 (0.71–1.52) |
| Monotherapy (adjusted) | 26/111,261 | 111/496,750 | 1.07 (0.70–1.66) |
| Negative control analysis (adjusted) | 14/107,576 | 194/721,192 | 0.48 (0.28–0.83) |

**Fig 2. Relative risks of any MCM and 42 selected individual MCMs (sorted by the most common to the least common MCM in the organ-specific groups) in the macrolide exposure group compared with the amoxicillin exposure group (continued).**

| Outcome | Macrolide group N events/N total | Amoxicillin group N events/N total | RR (95%CI) | |
|---|---|---|---|---|
| **Digestive system (cont.)** | | | | |
| **Diaphragmatic hernia** | | | | |
| Main analysis (crude) | 22/140,708 | 122/592,652 | 0.76 (0.48–1.20) | |
| Main analysis (adjusted) | 22/140,707 | 122/592,643 | 0.78 (0.49–1.25) | |
| Monotherapy (adjusted) | 20/111,261 | 98/496,750 | 0.90 (0.55–1.48) | |
| Negative control analysis (adjusted) | 11/107,576 | 166/721,192 | 0.46 (0.25–0.85) | |
| **Hirschrung's disease** | | | | |
| Main analysis (crude) | 18/140,708 | 66/592,652 | 1.15 (0.68–1.93) | |
| Main analysis (adjusted) | 18/140,707 | 66/592,643 | 1.15 (0.67–1.97) | |
| Monotherapy (adjusted) | 13/111,261 | 49/496,750 | 1.14 (0.61–2.13) | |
| Negative control analysis (adjusted) | 12/107,576 | 83/721,192 | 0.95 (0.52–1.74) | |
| **Atresia or stenosis of intestine** | | | | |
| Main analysis (crude) | 17/140,708 | 66/592,652 | 1.08 (0.64–1.85) | |
| Main analysis (adjusted) | 17/140,707 | 66/592,643 | 1.14 (0.66–1.97) | |
| Monotherapy (adjusted) | 13/111,261 | 58/496,750 | 1.05 (0.56–1.94) | |
| Negative control analysis (adjusted) | 14/107,576 | 73/721,192 | 1.37 (0.77–2.44) | |
| **Anomalies of intestinal fixation** | | | | |
| Main analysis (crude) | 14/140,708 | 54/592,652 | 1.09 (0.61–1.97) | |
| Main analysis (adjusted) | 14/140,707 | 54/592,643 | 1.03 (0.56–1.87) | |
| Monotherapy (adjusted) | 8/111,261 | 47/496,750 | 0.69 (0.32–1.47) | |
| Negative control analysis (adjusted) | 9/107,576 | 60/721,192 | 1.02 (0.51–2.07) | |
| **Abdominal wall defects** | | | | |
| **Omphalocele** | | | | |
| Main analysis (crude) | 21/140,708 | 92/592,652 | 0.96 (0.60–1.54) | |
| Main analysis (adjusted) | 21/140,707 | 92/592,643 | 0.97 (0.60–1.56) | |
| Monotherapy (adjusted) | 18/111,261 | 76/496,750 | 1.07 (0.64–1.81) | |
| Negative control analysis (adjusted) | 17/107,576 | 109/721,192 | 1.11 (0.67–1.85) | |
| **Gastroschisis** | | | | |
| Main analysis (crude) | 17/140,708 | 64/592,652 | 1.12 (0.66–1.91) | |
| Main analysis (adjusted) | 17/140,707 | 64/592,643 | 1.35 (0.79–2.31) | |
| Monotherapy (adjusted) | 12/111,261 | 56/496,750 | 1.15 (0.61–2.15) | |
| Negative control analysis (adjusted) | 18/107,576 | 98/721,192 | 1.19 (0.72–1.98) | |
| **Anomalies of kidney and urinary tract** | | | | |
| **Hydronephrosis** | | | | |
| Main analysis (crude) | 209/140,708 | 985/592,652 | 0.89 (0.77–1.04) | |
| Main analysis (adjusted) | 209/140,707 | 985/592,643 | 0.92 (0.79–1.07) | |
| Monotherapy (adjusted) | 165/111,261 | 816/496,750 | 0.92 (0.78–1.10) | |
| Negative control analysis (adjusted) | 165/107,576 | 1,069/721,192 | 1.04 (0.88–1.22) | |
| **Unilateral Renal Agenesis** | | | | |
| Main analysis (crude) | 47/140,708 | 223/592,652 | 0.89 (0.65–1.22) | |
| Main analysis (adjusted) | 47/140,707 | 223/592,643 | 0.93 (0.67–1.28) | |
| Monotherapy (adjusted) | 41/111,261 | 181/496,750 | 1.06 (0.75–1.50) | |
| Negative control analysis (adjusted) | 36/107,576 | 266/721,192 | 0.92 (0.65–1.31) | |
| **Renal Dysplasia** | | | | |
| Main analysis (crude) | 40/140,708 | 172/592,652 | 0.98 (0.69–1.38) | |
| Main analysis (adjusted) | 40/140,707 | 172/592,643 | 0.99 (0.70–1.41) | |
| Monotherapy (adjusted) | 36/111,261 | 152/496,750 | 1.08 (0.75–1.57) | |
| Negative control analysis (adjusted) | 26/107,576 | 186/721,192 | 0.95 (0.63–1.44) | |
| **Horseshoe kidney** | | | | |
| Main analysis (crude) | 26/140,708 | 146/592,652 | 0.75 (0.49–1.14) | |
| Main analysis (adjusted) | 26/140,707 | 146/592,643 | 0.76 (0.50–1.17) | |
| Monotherapy (adjusted) | 18/111,261 | 124/496,750 | 0.65 (0.39–1.07) | |
| Negative control analysis (adjusted) | 26/107,576 | 154/721,192 | 1.17 (0.77–1.78) | |
| **Posterior urethral valve** | | | | |
| Main analysis (crude) | 18/140,708 | 67/592,652 | 1.13 (0.67–1.90) | |
| Main analysis (adjusted) | 18/140,707 | 67/592,643 | 1.21 (0.71–2.05) | |
| Monotherapy (adjusted) | 11/111,261 | 54/496,750 | 0.93 (0.48–1.80) | |
| Negative control analysis (adjusted) | 12/107,576 | 95/721,192 | 0.85 (0.46–1.55) | |

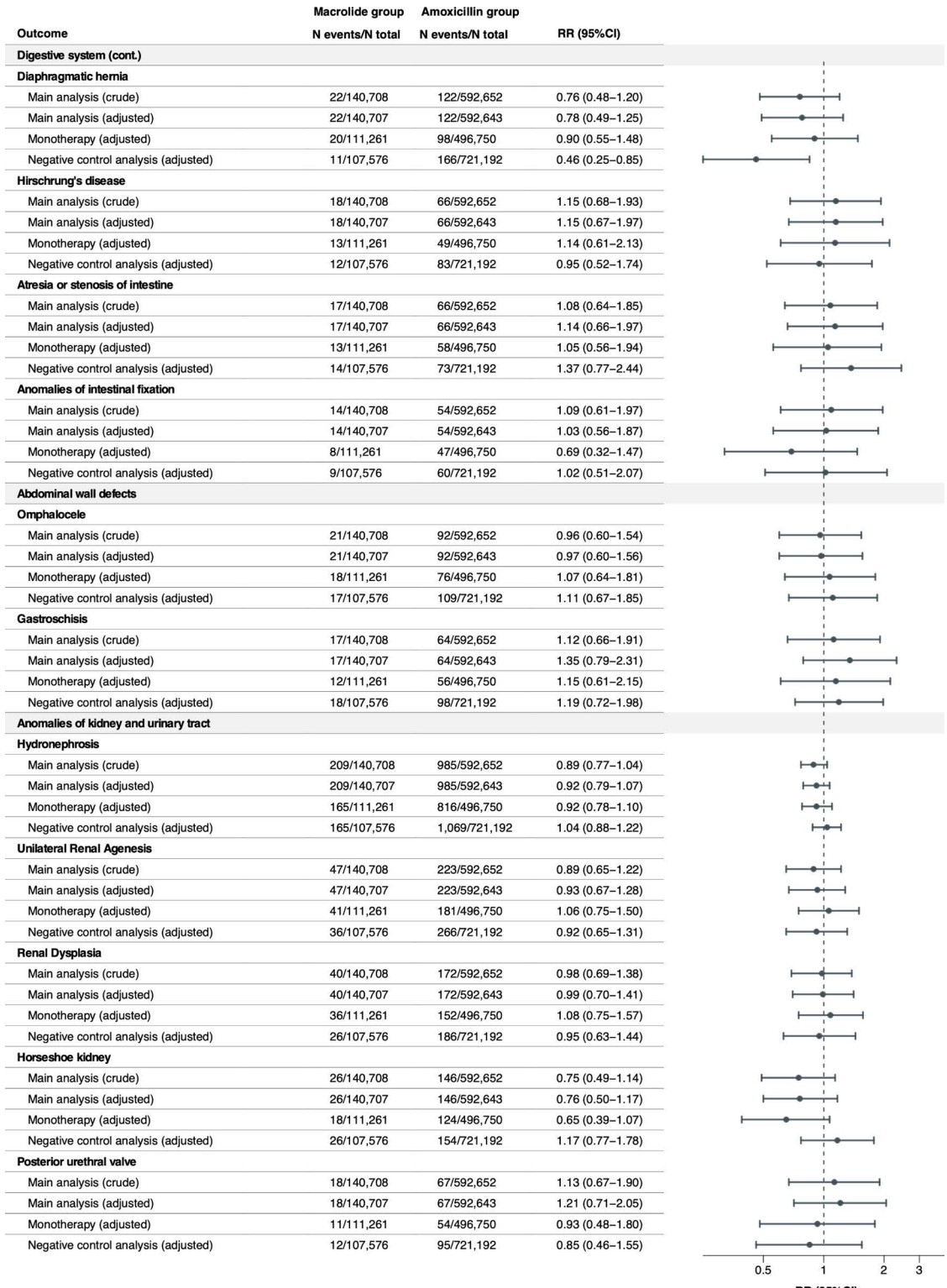

**Fig 3. Relative risks of any MCM and 42 selected individual MCMs (sorted by the most common to the least common MCM in the organ-specific groups) in the macrolide exposure group compared with the amoxicillin exposure group (continued).**

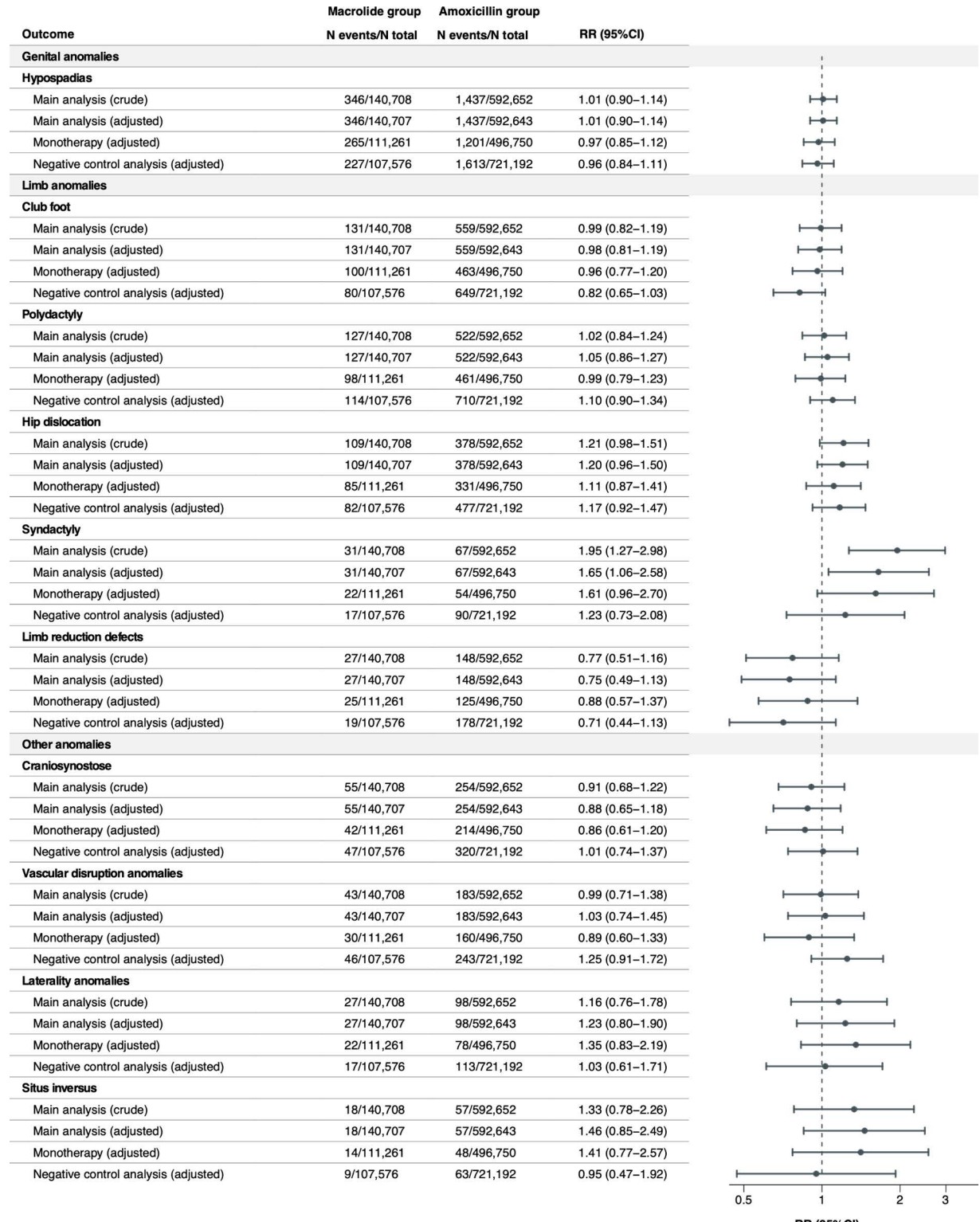

| Outcome | Macrolide group N events/N total | Amoxicillin group N events/N total | RR (95%CI) |
|---|---|---|---|
| **Genital anomalies** | | | |
| **Hypospadias** | | | |
| Main analysis (crude) | 346/140,708 | 1,437/592,652 | 1.01 (0.90–1.14) |
| Main analysis (adjusted) | 346/140,707 | 1,437/592,643 | 1.01 (0.90–1.14) |
| Monotherapy (adjusted) | 265/111,261 | 1,201/496,750 | 0.97 (0.85–1.12) |
| Negative control analysis (adjusted) | 227/107,576 | 1,613/721,192 | 0.96 (0.84–1.11) |
| **Limb anomalies** | | | |
| **Club foot** | | | |
| Main analysis (crude) | 131/140,708 | 559/592,652 | 0.99 (0.82–1.19) |
| Main analysis (adjusted) | 131/140,707 | 559/592,643 | 0.98 (0.81–1.19) |
| Monotherapy (adjusted) | 100/111,261 | 463/496,750 | 0.96 (0.77–1.20) |
| Negative control analysis (adjusted) | 80/107,576 | 649/721,192 | 0.82 (0.65–1.03) |
| **Polydactyly** | | | |
| Main analysis (crude) | 127/140,708 | 522/592,652 | 1.02 (0.84–1.24) |
| Main analysis (adjusted) | 127/140,707 | 522/592,643 | 1.05 (0.86–1.27) |
| Monotherapy (adjusted) | 98/111,261 | 461/496,750 | 0.99 (0.79–1.23) |
| Negative control analysis (adjusted) | 114/107,576 | 710/721,192 | 1.10 (0.90–1.34) |
| **Hip dislocation** | | | |
| Main analysis (crude) | 109/140,708 | 378/592,652 | 1.21 (0.98–1.51) |
| Main analysis (adjusted) | 109/140,707 | 378/592,643 | 1.20 (0.96–1.50) |
| Monotherapy (adjusted) | 85/111,261 | 331/496,750 | 1.11 (0.87–1.41) |
| Negative control analysis (adjusted) | 82/107,576 | 477/721,192 | 1.17 (0.92–1.47) |
| **Syndactyly** | | | |
| Main analysis (crude) | 31/140,708 | 67/592,652 | 1.95 (1.27–2.98) |
| Main analysis (adjusted) | 31/140,707 | 67/592,643 | 1.65 (1.06–2.58) |
| Monotherapy (adjusted) | 22/111,261 | 54/496,750 | 1.61 (0.96–2.70) |
| Negative control analysis (adjusted) | 17/107,576 | 90/721,192 | 1.23 (0.73–2.08) |
| **Limb reduction defects** | | | |
| Main analysis (crude) | 27/140,708 | 148/592,652 | 0.77 (0.51–1.16) |
| Main analysis (adjusted) | 27/140,707 | 148/592,643 | 0.75 (0.49–1.13) |
| Monotherapy (adjusted) | 25/111,261 | 125/496,750 | 0.88 (0.57–1.37) |
| Negative control analysis (adjusted) | 19/107,576 | 178/721,192 | 0.71 (0.44–1.13) |
| **Other anomalies** | | | |
| **Craniosynostose** | | | |
| Main analysis (crude) | 55/140,708 | 254/592,652 | 0.91 (0.68–1.22) |
| Main analysis (adjusted) | 55/140,707 | 254/592,643 | 0.88 (0.65–1.18) |
| Monotherapy (adjusted) | 42/111,261 | 214/496,750 | 0.86 (0.61–1.20) |
| Negative control analysis (adjusted) | 47/107,576 | 320/721,192 | 1.01 (0.74–1.37) |
| **Vascular disruption anomalies** | | | |
| Main analysis (crude) | 43/140,708 | 183/592,652 | 0.99 (0.71–1.38) |
| Main analysis (adjusted) | 43/140,707 | 183/592,643 | 1.03 (0.74–1.45) |
| Monotherapy (adjusted) | 30/111,261 | 160/496,750 | 0.89 (0.60–1.33) |
| Negative control analysis (adjusted) | 46/107,576 | 243/721,192 | 1.25 (0.91–1.72) |
| **Laterality anomalies** | | | |
| Main analysis (crude) | 27/140,708 | 98/592,652 | 1.16 (0.76–1.78) |
| Main analysis (adjusted) | 27/140,707 | 98/592,643 | 1.23 (0.80–1.90) |
| Monotherapy (adjusted) | 22/111,261 | 78/496,750 | 1.35 (0.83–2.19) |
| Negative control analysis (adjusted) | 17/107,576 | 113/721,192 | 1.03 (0.61–1.71) |
| **Situs inversus** | | | |
| Main analysis (crude) | 18/140,708 | 57/592,652 | 1.33 (0.78–2.26) |
| Main analysis (adjusted) | 18/140,707 | 57/592,643 | 1.46 (0.85–2.49) |
| Monotherapy (adjusted) | 14/111,261 | 48/496,750 | 1.41 (0.77–2.57) |
| Negative control analysis (adjusted) | 9/107,576 | 63/721,192 | 0.95 (0.47–1.92) |

RR (95%CI)

**Fig 4. Relative risks of any MCM and 42 selected individual MCMs (sorted by the most common to the least common MCM in the organ-specific groups) in the macrolide exposure group compared with the amoxicillin exposure group (continued).**

to amoxicillin in the EPI-MERES Register was 1.26 (95% CI 1.23 to 1.28) (S2 Fig). Finally, the *p*-values from 42 associations assessed in the main analysis are presented in S16 Table. After adjusting for multiple comparisons, none of the *p*-values remained statistically significant.

## Discussion

In this large population-based cohort study, we evaluated the risks of several individual MCMs among around 140,000 pregnancies exposed to macrolide antibiotics compared with over 592,000 pregnancies exposed to amoxicillin during the first trimester. While we did not find evidence for associations between exposure to macrolides and an increased risk for most MCMs, increased risks for spina bifida and syndactyly were observed. Sensitivity analyses yielded similar results to those in the main analysis. Negative control analysis revealed a null association for spina bifida and syndactyly. In the individual macrolide analyses, we consistently observed elevated RRs for these two malformations, however with large confidence intervals due to small numbers of events.

Previous observational studies have reported potential associations between macrolide exposure and cardiovascular malformations [25], genital malformations [10], digestive malformations [12,26,27], atrioventricular septal [25], erythromycin exposure and cardiovascular malformations [28,29], urinary system malformations [30], diaphragmatic hernia [31], limb reduction defects [32], and clarithromycin exposure and orofacial cleft [31]. However, the findings from our study did not support any of those associations. Conversely, increased risks of spina bifida and syndactyly following exposure to macrolides were observed. While associations were observed on a relative scale, the risk differences were relatively small. In the literature, data on the risk of individual MCMs were sparse. We found one case–control study mentioning a potential increased risk of syndactyly associated with exposure to azithromycin (adjusted odds ratio 3.80, 95% CI 1.62 to 8.94), but not other macrolides, based on 8 azithromycin-exposed cases [31]. To our knowledge, no prior study has found an association between macrolide exposure and an increased risk of spina bifida. However, the study of Fan and colleagues reported an elevated point estimate for nervous system malformations, albeit with a wide confidence interval (RR 2.29, 95% CI 0.95 to 5.55) [10]. A study by Cheng and colleagues recently demonstrated that unintentional exposure to high levels of macrolides in the environment was associated with an increased risk for neural tube defects [33]. Given that termination of pregnancy for spina bifida is common [34], prior studies probably could not include enough exposed pregnancies to be able to detect the signals among live births. There is some evidence from animal studies showing that prenatal exposure to certain macrolide antibiotics can cause fetal developmental toxicity [35]. To date, no established biological mechanism explains the observed associations with spina bifida and syndactyly. One proposed hypothesis is that macrolides can induce apoptosis [36], an important process in shaping tissues and organs during organogenesis. If macrolides induce apoptosis in certain fetal issues during the organogenesis period, it could disrupt normal fetal development and lead to MCMs.

Given the heterogeneity of pathophysiological mechanisms underlying multiple MCMs [37], it is crucial to investigate MCMs individually to avoid missing safety signals. As numerous MCMs were assessed concurrently in our study, concerns over multiple comparisons might arise. When applying the Benjamini–Hochberg correction, it was almost expected that none of the associations in the main analysis remained statistically significant considering the large number of analyses conducted and the limited number of macrolide-exposed events. Although we cannot entirely rule out the possibility that the increased risk for spina bifida and syndactyly was observed by chance, the elevated estimate points persisted across individual macrolide and sensitivity analyses. The findings from the negative control analysis were, as expected, null. Furthermore, an increased risk of syndactyly and neural tube defects associated

with macrolide exposure during the first trimester was observed in some previous studies [10,31,33]. Thus, these findings should be considered as safety signals warranting further investigation. Different signals of increased risks of other MCMs associated with individual macrolide exposures were also detected. However, non-informative results and very wide CIs from sensitivity analyses due to no or very few numbers of events prevented drawing conclusions.

## Strengths and limitations

The strengths of this study include its large, unselected cohort of pregnancies based on national healthcare databases, with the number of pregnancies exposed to macrolides more than 10 times higher than in the largest prior study by Andersson and colleagues (2021) [11]. This enabled us to assess several maternal characteristics and to investigate a wide range of MCMs separately. In our study, 5% of associations analyzed were expected to be statistically significant simply by chance. Although we acknowledge the possibility of false-positive findings, the *p*-value use could lead to missing potential safety signals [38]. Therefore, our interpretation was based on the association strength, the confidence interval width, the consistency of the results in sensitivity analyses, and the null negative control association. Given the exploratory nature of our study, we need to keep the observed associations under surveillance and reevaluate them in future research. Replicating our findings in different populations and studies is essential to establish more robust evidence.

Our study is subject to certain limitations due to constraints on its design. First, the study population was restricted to pregnancies resulting in live births, as information on MCMs was rarely available for pregnancies resulting in non-live birth outcomes. Since macrolide exposure was shown not to be strongly associated with live births, this bias is unlikely to affect our results. Second, antibiotic use during hospitalization is unavailable in the SNDS but is limited as antibiotics are mainly prescribed and started in primary care. Third, indications of antibiotic treatments are not available in the SNDS. To mitigate confounding by indication, we used pregnancies exposed to amoxicillin, an antibiotic sharing a similar antibacterial spectrum with macrolides, as the main active comparator group and cephalosporins as the second comparator. Fourth, there was potential residual confounding by unmeasured and poorly measured variables in the SNDS, such as maternal body mass index and lifestyle factors, which may be risk factors for birth defects [39,40]. However, the null findings in the negative control analysis showed that residual confounding tended to be minimized. Although non-reimbursed folic acid supplements are not captured in the SNDS, their use is probably not distributed differently between the macrolide and amoxicillin exposure groups. The non-differential misclassification bias might reduce the statistical power but is unlikely to affect markedly our interpretation. Fifth, we did not evaluate any dose–effect relationship as we could not determine the prescribed doses, the exact number of dispensed tablets, and the adherence of pregnant women to treatments.

In conclusion, our findings confirm that macrolide antibiotics are not associated with most individual MCMs. However, an increase in the risks of spina bifida and syndactyly cannot be ruled out, warranting ongoing monitoring and further investigation. In the meantime, macrolides should only be prescribed during the first trimester of pregnancy when necessary, and alternative antibiotics with a better-established safety profile should be considered whenever possible.

## Supporting information

**S1 Text. Presentation of the databases in the SNDS.**
(DOCX)

**S1 Table. Algorithms to identify teratogenic infections or suspected teratogenic infections.**
(DOCX)

**S2 Table. List of known teratogenic drugs.**
(DOCX)

**S3 Table. Algorithms for identifying major congenital malformations in the SNDS.**
(DOCX)

**S4 Table. Definition of covariates included in the propensity score model and their assessment periods.**
(DOCX)

**S5 Table. Description of antibiotic treatment during the first trimester and pregnancy outcome of the study cohort.**
(DOCX)

**S6 Table. Standardized differences (%) of pregnancies exposed to macrolides overall/each of the six macrolides and amoxicillin during the first trimester (before and after propensity score adjustment).**
(DOCX)

**S7 Table. Adjusted risk differences (per 10,000 live-born infants) of any MCM and 42 selected individual MCMs (sorted by the most common to the least common MCMs in the organ-specific groups) in pregnancies exposed to macrolides overall during the first trimester compared with amoxicillin.**
(DOCX)

**S8 Table. Adjusted relative risks of any MCM and 42 selected individual MCMs (sorted by the most common to the least common MCMs in the organ-specific groups) in pregnancies exposed to macrolides overall compared with amoxicillin during a narrower exposure window.**
(DOCX)

**S9 Table. Adjusted relative risks of any MCM and 42 selected individual MCMs (sorted by the most common to the least common MCMs) in pregnancies exposed to each of the six macrolides compared with amoxicillin: results from the main analysis.**
(DOCX)

**S10 Table. Relative risks of any MCM and 42 selected individual MCMs (sorted by the most common to the least common MCMs in the organ-specific groups) in pregnancies exposed to each of six individual macrolides during the first trimester compared with amoxicillin: results from main, sensitivity, and negative control analyses.**
(DOCX)

**S11 Table. Supplementary analysis - Adjusted relative risks of any MCM and 42 selected individual MCMs (sorted by the most common to the least common MCMs in the organ-specific groups) in pregnancies exposed to macrolides overall during the first trimester compared with the cephalosporin exposure group.**
(DOCX)

**S12 Table. Supplementary analysis - Adjusted relative risks of any MCM and 42 selected individual MCMs (sorted by the most common to the least common MCMs in the**

organ-specific groups) in pregnancies exposed to macrolides overall during the first trimester compared with the unexposed group.
(DOCX)

**S13 Table. Supplementary analysis - Adjusted relative risks of 12 organ-specific MCM groups in pregnancies exposed to macrolides during the first trimester compared with amoxicillin.**
(DOCX)

**S14 Table. Post-hoc analysis - Adjusted relative risks of spina bifida and syndactyly across baseline characteristic strata.**
(DOCX)

**S15 Table. Post-hoc analysis - Results from the main analysis when applying propensity score overlap weights.**
(DOCX)

**S16 Table. Post-hoc analysis – Statistical significance of results from the main analyses with the Benjamini-Hochberg (BH) procedure (5% false positive rate).**
(DOCX)

**S1 Fig. Flowchart of the study cohort.**
(DOCX)

**S2 Fig. Flowchart for comparing the frequency of non-live births among pregnancies exposed to macrolides during the first trimester with those exposed to amoxicillin in the EPI-MERES Register.**
(DOCX)

**S1 Checklist. STROBE, Strengthening the Reporting of Observational Studies in Epidemiology.**
(DOCX)

**S1 Protocol. Summary protocol.**
(DOCX)

## Author contributions

**Conceptualization:** Anh Tran, Mahmoud Zureik, Xavier Duval, Sarah Tubiana.

**Data curation:** Jérôme Drouin.

**Formal analysis:** Anh Tran.

**Supervision:** Mahmoud Zureik, Xavier Duval, Sarah Tubiana.

**Writing – original draft:** Anh Tran.

**Writing – review & editing:** Mahmoud Zureik, Jeanne Sibiude, Sara Miranda, Lise Marty, Alain Weill, Rosemary Dray-Spira, Xavier Duval, Sarah Tubiana.

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
