## [Editor Report · Decision Letter 0]

11 Oct 2024

Dear Dr Tran, 

Thank you for submitting your manuscript entitled "Exposure to macrolide antibiotics during the first trimester of pregnancy and risk of major congenital malformations: A French nationwide cohort study" for consideration by PLOS Medicine.

Your manuscript has now been evaluated by the PLOS Medicine editorial staff and I am writing to let you know that we would like to send your submission out for external peer review.

Please re-submit your manuscript within two working days, i.e. by Oct 15 2024 11:59PM.

Feel free to email me at ssunny@plos.org@plos.org if you have any queries relating to your submission.

Kind regards,

Syba Sunny, MBBS, MRes, FRCPath

Senior Editor

PLOS Medicine

---

## [Editor Report · Decision Letter 1]

14 Oct 2024

Dear Dr Tran, 

Thank you for submitting your manuscript entitled "Exposure to macrolide antibiotics during the first trimester of pregnancy and risk of major congenital malformations: A French nationwide cohort study" for consideration by PLOS Medicine.

Your manuscript has now been evaluated by the PLOS Medicine editorial staff and I am writing to let you know that we would like to send your submission out for external peer review.

Please re-submit your manuscript within two working days, i.e. by Oct 16 2024 11:59PM. Please do let us know if you need more time.

Kind regards,

Syba

Syba Sunny, MBBS, MRes, FRCPath

Associate Editor

PLOS Medicine

---

## [Decision Letter · Decision Letter 2]

22 Nov 2024

Dear Dr Tran,

Many thanks for submitting your manuscript "Exposure to macrolide antibiotics during the first trimester of pregnancy and risk of major congenital malformations: A French nationwide cohort study" (PMEDICINE-D-24-03341R2) to PLOS Medicine. The paper has been reviewed by subject experts and a statistician; their comments are included below and can also be accessed here: [LINK]

As you will see, the reviewers were generally quite positive about your manuscript but there were several calls for clarification/extra information and suggestions for improvement. After discussing the paper with the editorial team and an academic editor with relevant expertise, I'm pleased to invite you to revise the paper in response to the reviewers' comments. We plan to send the revised paper to some or all of the original reviewers, and we cannot provide any guarantees at this stage regarding publication.

We ask that you submit your revision by Dec 13 2024 11:59PM. However, if this deadline is not feasible, please contact me by email, and we can discuss a suitable alternative.

Don't hesitate to contact me directly with any questions (ssunny@plos.org). 

Best regards, 

Syba 

Syba Sunny, MBBS, MRes, FRCPath 

Associate Editor

PLOS Medicine

ssunny@plos.org

Comments from the academic editor:

The academic editor was supportive of your manuscript moving forward to a major revision outcome. However, he also made the following comments, that he asks the authors to respond to:

- Given that macrolides were tested against many different outcomes, it seems fairly likely that a statistically significant result could occur due to random chance. In this circumstance, it seems essential that results that are adjusted for multiple comparison also be presented (even if original estimates are also maintained).

- Another way to evaluate the robustness of the results would be to assess with association between macrolides and spinal bifida/syndactyly hold up in stratified analyses. One would expect the effect size would be similar across characteristics strata presented in Table 1 if these association are indeed real. This could be presented as a supplement.

Comments from the reviewers: 

Reviewer #1: This study utilizes data from the Mother-Child EPI-MERES Register nested within the French Health Data System (SNDS) to conduct a large population-based cohort study. The findings indicate that the use of macrolide antibiotics by mothers during the first trimester does not have a strong teratogenic effect, but may increase the risk of spina bifida and syndactyly in offspring. This study has an adequate sample size, a reasonable research design, rigorous statistical analysis, fluent text expression, and interesting results that are of guiding significance for clinical maternal and child health care. I suggest that this study be published after addressing the following issues:

1. This study focuses on the impact of macrolide antibiotic use in the first trimester (from the first day of LMP to the end of the 12th gestational week) on major congenital malformations in offspring. However, considering that antibiotic exposure during the periconceptional period (shortly before pregnancy) may have residues in the body, how can the impact of this situation be excluded?

2. The study defines antibiotic exposure as oral macrolide use. How should the impact of other exposure routes, such as dermal intake, be considered?

3. Given the evaluation of the effects of multiple antibiotics on various subtypes of birth defects, why did the authors not employ multiple testing analysis to correct the statistical results?

4. It is recommended to add a discussion on the possible biological mechanisms of macrolide antibiotics leading to spina bifida or syndactyly in the discussion section.

Reviewer #2: The authors present a large body of work exploring the association between macrolide exposure in early pregnancy and major congenital malformations, compared with amoxicillin exposure. The study has been well designed and conducted with several sensitivity analyses to explore the robustness of the findings. 

Regarding the statistical approach, the authors have utilised a propensity score based fine stratification method and used 50 strata. These methods are appropriate, and they show balance achieved for the included covariates. Specific comments:

- Did the authors consider alternate models to achieve further balance rather than trimming? 

- Additionally, how were covariates selected for the models? Were DAGs considered for confounders? 

- Given both control and exposed groups are antibiotic exposed they are likely similar and few variables would be considered confounders. 

- Missing data - what proportion of data were missing and how was this handled?

Other considerations: 

Title: consider adding mention to the comparison to amoxicillin - this better reflects the actual study. 

Abstract: include the overall RR in the abstract and 95% CI. Line 46 of the conclusion should be amended to association and not effect - this is a population based study. Consider 'are not strongly associated with'.

Introduction: objective should be updated to include amoxicillin as - ie exposure to macrolides compared with amoxicillin to better reflect the study.

Results: 

-Why is maternal age highlighted by exposure groups but not other covariates? Further discussion on how the two groups differ is needed. 

-Include the percent of each macrolide rather than only n. 

-Line 200 - is this trying to get to suggest nulliparity? May be better to say that it was the proportion of women with only one live birth in the dataset or nulliparous based on data from 2010 - 2020. 

-Needs to clarified whether prevalence per 10,000 is based on raw data or adjusted throughout. 

- Suggest including the unadjusted relative risk for the overall results and adjusted - this gives the reader an idea of the impact of confounding

-Line 221 consider comparing the prevalence statistically and whether this is a clinically meaningful difference 

-line 226: include a summary of these supplemental findings, they show the robustness of the findings. 

- Replace 'RR estimates'; could be along the lines of the association between /the adjusted relative risk etc 

-Line 235 - needs more detail, a broad statement for a result - did the 95% CIs cross 1? Difficult to draw conclusions from this. 

-A major consideration is that folic acid is often available over the counter, thus if this is obtained via linkage there will be many women in the control group who are indeed taking folic acid - introducing significant bias into this analysis. Is the point of this analysis to show that folic acid may attenuate this association? If so, the lack of accurate exposure is a major limitation and the authors should consider removing this analysis. 

- Live birth bias analysis is discussed in discussion but not results - should be added if discussed and were any statistical analysis done on this to compare the difference?

-BMI: Is there no information on BMI? The obesity related conditions will only capture women hospitalised but not overall BMI, this is an important maternal characteristic in the setting of anomalies. If this data is not available it should be acknowledged. 

Tables: Pregnancy related conditions - parity and folic acid are not conditions - adjust titles

Table 2 - are these unadjusted prevalences? If so this needs to be made clear. 

Table S10 - describes compared with the unexposed group but the table still shows amoxicillin - not total unexposed. 

Figure 1: consider including the unadjusted relative risk to clearly show the impact of confounding/adjustment on the results. Additionally, including the overall major congenital malformation risk at the top of the table is useful. 

Figure 2 is difficult to interpret. Why not include the 95% CIs so the reader can draw their conclusions off this data. The figure is large, considered removing or condensing.

Reviewer #3: Review Comments:

Thank you for the opportunity to review this manuscript. The authors present an interesting study to understand the association between macrolide antibiotics, which were commonly used in pregnancy, and major congenital malformations using the French National Health System database. Overall, this study is well-designed, but I would like to suggest a few clarifications and suggestions to further strengthen the manuscript.

Introduction

1. Line 66: Please mention the types of studies included in the meta-analysis (eg, case-control, cohort design) and their pooled sample sizes in the main text.

Method - Data sources

1. Line 81: Can you provide a bit further details on the database used? For example, please provide an essential explanation of the SNDS (eg, the type of data source) or provide relevant references. Regarding the EPI-MERES register, please provide simple linkage methods and linkage % from the total population. As for the date of conception, can you briefly describe how this information is collected in this database?

2. Line 90: Could you elaborate on "Data for same-sex twins, however, are not included due to technical constraints."? Does that mean that you cannot distinguish between the twins in the data? Or we cannot see the record of the outcome at all?

Method - Study population

1. Line 104: Is there any reason why the patients were restricted to those who had at least one record in the SDND in the 2 preceding years? 

2. Line 106: Could you clarify the definition or codes for chromosomal abnormality?

3. Line 109-111: Is there any reference about "Pregnancies with three or more prescriptions filled for spiramycin between the date of conception and the date of delivery were classified as having Toxoplasmosis and excluded at this stage."?

Method - Exposure

1. Line 116: Is there any reason or rationale for including only oral formulations while excluding injectables?

2. Line 119: Does the 'index date' mean the start date of follow-up? If not, please avoid using the 'index date' term. If any, please clarify the start and end dates of follow-up as well as censoring criteria in the Method section.

Method - Outcomes 

1. Potential outcome misclassification: Is there any validation study or positive predictive values (PPVs) for the outcome (MCMs)? 

2. Did you define a minimum follow-up period, such as at least one or two years of follow-up to assess the outcome (except for censoring criteria)? For example, the inclusion period would be 2010-2020 and the follow-up period would be up to 2022. Or were infants with a follow-up time of only one day included as well? Please clarify the end dates of follow-up (or study period) in this study.

Method - Covariates

1. Please clarify the measurement window for some covariates, such as lifestyle factors.

2. As for pre-existing conditions:

1) Hypertension was defined based on antihypertensive drug prescriptions. However, these drugs have many other indications than hypertension (eg, migraine, anxiety, heart failure, etc). How about using the term 'antihypertensive drug use' rather than 'hypertension'?

2) As for obesity, I doubt the accuracy of the E66 code, as obesity is not usually well-recorded in the claims database. I suggest at least using the term "clinical visits due to obesity".

3) As for diabetes, some antidiabetic drugs can have other indications (such as obesity, and PCOS). I suggest using the term "antidiabetic drug use" rather than diabetes.

3. Line 137-9: "Gestational week when antibiotic treatment was started was also considered for the analysis between macrolide and amoxicillin groups." Please clarify how the gestational weeks exposed were handled from a statistical or methodological perspective.

Method - Statistical analysis

1. Line 156: Could you clarify why generalized estimating equations were used? Is it to consider any correlation due to more than one delivery episode in the same woman?

2. Line 157-8: Could you explain more about the sentence "No adjustments were made for multiple testing."?

Method - Sensitivity and supplementary analysis

1. Line 165-7: Could you clarify the rationale of the first sensitivity analysis? Is it to exclude the possible influence of other ingredients and purely look at the effect of the specific ingredient?

2. Line 177-9: How accurate is the data on maternal folic acid supplementation in the SNDS? I am curious because supplements are available privately or as over-the-counter drugs and often are not recorded in claims data.

Results/Discussion

1. Line 199-201 & Table 1: "Parity" refers to the number of times a woman has given birth to a fetus with >20 or 24 weeks of gestational age, regardless of whether the infant was born alive or stillborn. In Table 1, does the "Parity" mean that the first birth since 2010? If then, because of left censoring in the data, I don't think the variable is actual nulliparity (that is, a woman who has never given birth before 2010) and the variable would be not really meaningful.

2. Table 1: In France, are all assisted reproduction (oocyte pick-up, embryo transfer, and artificial insemination) reimbursed? Would this variable have a low probability of misclassification?

3. Can you assess any dose or duration-response relationship if information on the dose or duration of drug prescription is available?

3. Figure 1: The figure is very long, and it is not easy to directly identify the organ system associated with each outcome. I suggest that the outcome should be first classified by the organ system like Table 2, and then sorted by the most common to the least common MCMs in the organ system.

4. Figure 2: The figure focuses on the upper limit of 95% CI, but I suggest presenting both lower and upper limits because some estimates should have very wide CIs, which means that the risk can be observed by chance. This figure is good, but it is difficult to the statistical significance.

5. Line 293: Is there any reason to refer only to Andersson et al. in this sentence? Is this study the largest existing prior study?

---

* Please upload any figures associated with your paper as individual TIF or EPS files with 300dpi resolution at resubmission; please read our figure guidelines for more information on our requirements: http://journals.plos.org/plosmedicine/s/figures. While revising your submission, please upload your figure files to the PACE digital diagnostic tool, https://pacev2.apexcovantage.com/. PACE helps ensure that figures meet PLOS requirements. To use PACE, you must first register as a user. Then, login and navigate to the UPLOAD tab, where you will find detailed instructions on how to use the tool. If you encounter any issues or have any questions when using PACE, please email us at PLOSMedicine@plos.org.

* Thank you for providing a Data Availability statement. For data residing with a third party, authors are required to provide instructions with contact information (web or email address) for obtaining the data. Please note that a study author cannot be the contact person for the data. PLOS journals do not allow statements supported by "data not shown" or "unpublished results." For such statements, authors must provide supporting data or cite public sources that include it.

* I see that you have followed the STROBE reporting guideline. Please include the completed checklist as Supporting Information. When completing the checklist, please use section and paragraph numbers, rather than page numbers. Please add the following statement, or similar, to the Methods: "This study is reported as per the Strengthening the Reporting of Observational Studies in Epidemiology (STROBE) guideline (S1 Checklist)”, i.e. with an indication of the document’s location.

* When reporting 95% CIs please separate upper and lower bounds with commas instead of hyphens as the latter can be confused with reporting of negative values. 

* Please include page numbers (as well as line numbers) in the manuscript file. 

FIGURES AND TABLES

SUPPLEMENTARY MATERIAL

* Please note that supplementary material will be posted as supplied by the authors. 

OBSERVATIONAL STUDIES

* For all observational studies, in the manuscript text, please indicate: (1) the specific hypotheses you intended to test, (2) the analytical methods by which you planned to test them, (3) the analyses you actually performed, and (4) when reported analyses differ from those that were planned, transparent explanations for differences that affect the reliability of the study's results. If a reported analysis was performed based on an interesting but unanticipated pattern in the data, please be clear that the analysis was data driven. 

* Please state in the Methods section whether the study had a prospective protocol or analysis plan. If a prospective analysis plan (from your funding proposal, IRB or other ethics committee submission, study protocol, or other planning document written before analyzing the data) was used in designing the study, please include the relevant document(s) with your revised manuscript as a Supporting Information file to be published alongside your study and cite it in the Methods section. A legend for this file should be included at the end of your manuscript. If no such document exists, please make sure that the Methods section transparently describes when analyses were planned, and when/why any data-driven changes to analyses took place. Changes in the analysis, including those made in response to peer review comments, should be identified as such in the Methods section of the paper, with rationale.

---

## [Decision Letter · Decision Letter 3]

24 Jan 2025

Dear Dr Tran,

Many thanks for submitting your manuscript "First-trimester exposure to macrolides and risk of major congenital malformations compared with amoxicillin: A French nationwide cohort study" (PMEDICINE-D-24-03341R3) to PLOS Medicine. The paper has been reviewed by subject experts and a statistician; their comments are included below and can also be accessed here: [LINK]

As you will see, the subject matter expert reviewers were happy with your revisions. However, the statistician had some more requests for the authors to address. In addition, the academic editor had a couple of requests. Following a discussion amongst the in-house editors, and given the time that we predict would be required to address these additional requests, we have decided to issue another major revision decision. We plan to send your next revision to the statistician and the academic editor, and we cannot provide any guarantees at this stage regarding publication.

We ask that you submit your revision by Feb 14 2025 11:59PM. However, if this deadline is not feasible, please contact me by email, and we can discuss a suitable alternative.

Best regards, 

Syba 

Syba Sunny, MBBS, MRes, FRCPath 

Associate Editor

PLOS Medicine

ssunny@plos.org

Comments from the academic editor:

The academic editor supported the decision to continue engaging with the paper, but asked that the authors address 2 specific requests. 

(1) He asked that the authors include the analysis that was adjusted for multiple comparisons more formally in the Methods section and discuss this in the Results. 

(2) He also asked that the authors incorporate this analysis into the authors’ interpretation of the results in the Discussion (not just in the limitations section).

Comments from the reviewers: 

Reviewer #1: Reviews on manuscript PMEDICINE-D-24-03341R3

Upon this second review, I'm pleased to note that the authors have effectively addressed all the concerns I raised in my initial review. The authors have carefully revised the manuscript, providing clear explanations and relevant data. Therefore� I recommend this manuscript for publication in the journal.

Reviewer #2: I thank the authors for their considered replies and attempt to address all queries, they should be commended on this effort. I have four further queries: 

1. Were all covaries listed in the 'covariates' heading included in the models? It is unclear how this is currently worded with 'considered'. 

2. How was gestational age at exposure handled? This is an important difference that was not clear earlier. It is plausible that the earlier exposure window for macrolides compared with amoxicillin (approx. 1 week earlier) may influence the outcome (greatest risk of neural tube defects occurring approximately weeks 3 -4). 

3. The authors refer to the Benjamini-Hochberg correction, which although appropriate given the number of hypotheses tested, this is not seen in the manuscript (only discussion). Details of this should be included, including what the q-value was set at and results of this (as a supplement). 

4. The potential limitation of misclassification of folic acid use due to only prescriptions captured needs to be further highlighted. 

Reviewer #3: I sincerely appreciate the dedicated efforts of the authors in addressing my comments and the thoughtful guidance provided by the editors. All my comments have been thoroughly addressed, and I am grateful for the opportunity to contribute to this excellent work.

---

Editorial requests:

* Please ensure that the paper adheres to the PLOS Data Availability Policy (see http://journals.plos.org/plosmedicine/s/data-availability). For data residing with a third party, authors are required to provide instructions with contact information (web or email address) for obtaining the data. Please note that a study author cannot be the contact person for the data. 

* It appears that the STROBE checklist was not uploaded in this revision. Could the authors make sure this is uploaded for the next revision please? Please also ensure that the checklist has section and paragraph numbers in the right hand column, rather than page numbers.

* Thank you for providing an Author Summary. The purpose of this section is to make your findings accessible to a wider audience. As such, some terms, though they may seem obvious to a specialist, may need some explanation. For example, please explain that macrolides are a class of antibiotics (in the first sentence). It may also be beneficial to (very briefly) explain why macrolides were compared to amoxicillin.

---

## [Decision Letter · Decision Letter 4]

10 Mar 2025

Dear Dr. Tran,

Thank you very much for re-submitting your manuscript "First-trimester exposure to macrolides and risk of major congenital malformations compared with amoxicillin: A French nationwide cohort study" (PMEDICINE-D-24-03341R4) for review by PLOS Medicine.

I have discussed the paper with my colleagues and the academic editor and it was also seen again by one reviewer. I am pleased to say that provided the remaining editorial and production issues are dealt with we are planning to accept the paper for publication in the journal.

[LINK]

We look forward to receiving the revised manuscript by Mar 17 2025 11:59PM.   

Sincerely,

Rebecca Kirk

On behalf of:

Suzanne De Bruijn, PhD

Senior Editor 

PLOS Medicine

plosmedicine.org

Requests from Editors:

GENERAL EDITORIAL REQUESTS

* Please confirm that your title complies with PLOS Medicine's style. Your title must be nondeclarative and not a question. It should begin with main concept if possible. "Effect of" or "risk of" should be used only if causality can be inferred, i.e., for an RCT. Please place the study design ("A randomized controlled trial," "A retrospective study," "A modelling study," etc.) in the subtitle (ie, after a colon).

* Please confirm that your abstract complies with our requirements, including providing all the information relevant to this study type https://journals.plos.org/plosmedicine/s/submission-guidelines#loc-abstract

* Please ensure that all abbreviations are defined at first use throughout the text.

* Please use a comma to separate the confidence intervals as using a 'dash' can be confused with a minus sign.

FUNDING STATEMENT

* The funding statement should include: specific grant numbers, initials of authors who received each award, URLs to sponsors’ websites. Also, please state whether any sponsors or funders (other than the named authors) played any role in study design, data collection and analysis, the decision to publish, or preparation of the manuscript. If they had no role in the research, include this sentence: “The funders had no role in study design, data collection and analysis, decision to publish, or preparation of the manuscript.”

COMPETING INTERESTS STATEMENT

* All authors must declare their relevant competing interests per the PLOS policy, which can be seen here: https://journals.plos.org/plosmedicine/s/competing-interests For authors with ties to industry, please indicate whether any of the interests has a financial stake in the results of the current study.

OBSERVATIONAL, COHORT, CROSS-SECTIONAL, AND CASE CONTROL STUDIES

* Did your study have a prospective protocol or analysis plan? Please state this (either way) early in the Methods section.

c) In either case, changes in the analysis-- including those made in response to peer review comments-- should be identified as such in the Methods section of the paper, with rationale."

* Your study is observational and therefore causality or calculation of risk cannot be inferred. Please remove language that implies causality and refer to associations instead.

Comments from Reviewers:

Reviewer #2: I thank the authors for their considered response and addressing these queries.

[LINK]

---

## [Editor Report · Decision Letter 5]

17 Mar 2025

Dear Dr Tran, 

On behalf of my colleagues and the Academic Editor, Aaloke Mody, I am pleased to inform you that we have agreed to publish your manuscript "First-trimester exposure to macrolides and risk of major congenital malformations compared with amoxicillin: A French nationwide cohort study" (PMEDICINE-D-24-03341R5) in PLOS Medicine.

PRESS

Sincerely, 

Rebecca Kirk 

Senior Editor 

PLOS Medicine